# Hierarchical Hybrid Sliced Wasserstein: A Scalable Metric for Heterogeneous Joint Distributions

**Khai Nguyen**
Department of Statistics and Data Sciences
The University of Texas at Austin
Austin, TX 78712
khainb@utexas.edu

**Nhat Ho**
Department of Statistics and Data Sciences
The University of Texas at Austin
Austin, TX 78712
minhnhat@utexas.edu

## Abstract

Sliced Wasserstein (SW) and Generalized Sliced Wasserstein (GSW) have been widely used in applications due to their computational and statistical scalability. However, the SW and the GSW are only defined between distributions supported on a homogeneous domain. This limitation prevents their usage in applications with heterogeneous joint distributions with marginal distributions supported on multiple different domains. Using SW and GSW directly on the joint domains cannot make a meaningful comparison since their homogeneous slicing operator, i.e., Radon Transform (RT) and Generalized Radon Transform (GRT) are not expressive enough to capture the structure of the joint supports set. To address the issue, we propose two new slicing operators, i.e., Partial Generalized Radon Transform (PGRT) and Hierarchical Hybrid Radon Transform (HHRT). In greater detail, PGRT is the generalization of Partial Radon Transform (PRT), which transforms a subset of function arguments non-linearly while HHRT is the composition of PRT and multiple domain-specific PGRT on marginal domain arguments. By using HHRT, we extend the SW into Hierarchical Hybrid Sliced Wasserstein (H2SW) distance which is designed specifically for comparing heterogeneous joint distributions. We then discuss the topological, statistical, and computational properties of H2SW. Finally, we demonstrate the favorable performance of H2SW in 3D mesh deformation, deep 3D mesh autoencoders, and datasets comparison[1].

## 1 Introduction

Optimal transport [55, 47] is a powerful mathematical tool for machine learning, statistics, and data sciences. As an example, Wasserstein distance [47], defined as the optimal transportation cost between two distributions, has been used successfully in many areas of machine learning and statistics, such as generative modeling on images [2, 53], representation learning [35], vocabulary learning [57], and so on. Despite being accepted as an effective distance, Wasserstein distance has been widely known as a computationally expensive distance. In particular, when comparing two distributions that have at most $n$ supports, the time complexity and the memory complexity of the Wasserstein distance scale with the order of $\mathcal{O}(n^3 \log n)$ [45] and $\mathcal{O}(n^2)$ respectively. In addition, the Wasserstein distance requires more samples to approximate a continuous distribution with its empirical distribution in high dimension since its sample complexity is of the order of $\mathcal{O}(n^{-1/d})$ [20], where $n$ is the sample size and $d$ is the number of dimensions. Therefore, Wasserstein distance is not statistically and computationally scalable, especially in high dimensions.

Along with entropic regularization [17] which can reduce the time complexity and memory complexity of computing optimal transport to $\mathcal{O}(n^2)$ and $\mathcal{O}(n^2)$ in turn, sliced Wasserstein (SW) distance [11]

---

[1]Code for this paper is published at https://github.com/khainb/H2SW.

is one alternative approach for the original Wasserstein distance. The key benefit of the SW distance is that it scales the order $\mathcal{O}(n \log n)$ and $\mathcal{O}(n)$ in terms of time and memory respectively. The reason behind that fast computation is the closed-form solution of optimal transport in one dimension. To leverage that closed-form, sliced Wasserstein utilizes Radon Transform [23] (RT) to transform a high-dimensional distribution to its one-dimensional projected distributions, then the final distance is calculated as the average of all one-dimensional Wasserstein distance. By doing that, the SW distance has a very fast sample complexity i.e., $\mathcal{O}(n^{-1/2})$, which makes it computationally and statistically scalable in any dimension. Therefore, the SW distance has been applied successfully in various domains of applications including generative models [19], domain adaptation [32], clustering [27], 3D shapes [30, 29], gradient flows [34, 8], Bayesian inference computation [37, 58], texture synthesis [22], and many other tasks.

Despite being useful, the SW distance is not as flexible as the Wasserstein distance in terms of choosing the ground metric. In greater detail, the number of ground metrics in one dimension is limited, especially ground metrics that lead to the closed-form solution. As a result, the role of capturing the structure of distributions belongs to the slicing/projecting operators. To generalize RT to non-linear projection, generalized Radon Transform (GRT) is introduced in [3] with circular projection [28], polynomial projection [51], and so on. With GRT, Generalized Sliced Wasserstein (GSW) distance is proposed in [26]. In addition, there is a line of works on developing sliced Wasserstein variants on different manifolds such as hyper-sphere [6, 54, 49, 50], hyperbolic manifolds [7], the manifold of symmetric positive definite matrices [10], general manifolds and graphs [52]. In those works, special variants of GRT are proposed.

Although the SW has become more effective on multiple domains, no SW variant is designed specifically for heterogeneous joint distributions i.e., joint distributions that have marginals supported on different domains, except for the product of Hadamard manifolds [9]. It is worth noting that marginal domains of heterogeneous joint distributions can be any metric space and are not necessary manifolds. Heterogeneous joint distributions appear in many applications, e.g., domain adaptation domains [15, 4], comparing datasets with labels [1], 3D shape deformation [29], and so on. In this case, Wasserstein distance can be adapted by using a mixed ground metric, i.e., a weighted sum of metrics on domains [15, 1]. In contrast to the Wasserstein distance, the adaptation of SW has not been well-investigated. Using GSW directly with one type defining function for all marginals cannot separate the information within and among groups of arguments.

**Contribution:** In this work, we tackle the challenge of designing a sliced Wasserstein variant for heterogeneous joint distributions. In summary, our main contributions are three-fold:

1. We first extend the partial Radon Transform to the partial generalized Radon Transform (PGRT) to inject non-linearity into local transformation. We discuss the injectivity of PGRT for some choices of defining functions. We then propose a novel slicing operator for heterogeneous joint distributions, named Hierarchical Hybrid Radon Transform (HHRT). In particular, HHRT is a hierarchical transformation that first applies partial generalized Radon Transform with different defining functions on arguments of each marginal to gather marginal information, then applies partial Radon Transform on the joint transformed arguments to gather information among marginals. We show that HHRT is injective as long as the partial generalized Radon Transform is injective.

2. We propose Hierarchical Hybrid Sliced Wasserstein (H2SW) which is a novel metric for comparing heterogeneous joint distributions by utilizing the HHRT. Moreover, we investigate the topological properties, statistical properties, and computational properties of H2SW. In particular, we show that H2SW is a valid metric on the space of distribution over the joint space, H2SW does not suffer from the curse of dimensionality and enjoys the same computational scalability as SW distance.

3. A 3D mesh can be effectively represented by a point-cloud and corresponding surface normal vectors. Therefore, it can be seen as an empirical heterogeneous joint distribution. We conduct experiments on optimization-based 3D mesh deformation and deep 3D mesh autoencoder to show the favorable performance of H2SW compared to SW and GSW. Moreover, we also illustrate that H2SW can also provide a meaningful comparison for probability distributions on the product of Hadamard manifolds by conducting experiments on dataset comparison.

**Organization.** We first provide some preliminaries on SW distance, GSW distance, and joint Wasserstein distance in Section 2. We then define the hierarchical hybrid Radon transform and hierarchical hybrid sliced Wasserstein distance s in Section 3. Section 4 contains experiments on 3D mesh deformation, deep 3D mesh autoencoder, and datasets comparison. We conclude the paper in Section 5. Finally, we defer the proofs of key results, and additional materials in the Appendices.

## 2 Preliminaries

**Wasserstein distance.** For $p \geq 1$, the Wasserstein-$p$ distance [55, 47] between two distributions $\mu \in \mathcal{P}(\mathcal{X})$ and $\nu \in \mathcal{P}(\mathcal{Y})$, where $\mathcal{X}$ and $\mathcal{Y}$ are subsets of $\mathbb{R}^d$ and they share a ground metric $c : \mathcal{X} \times \mathcal{Y} \to \mathbb{R}^+$, is defined as:

$$\mathrm{W}_p^p(\mu, \nu; c) := \inf_{\pi \in \Pi(\mu,\nu)} \int_{\mathcal{X} \times \mathcal{Y}} c(x, y)^p d\pi(x, y), \tag{1}$$

where $\Pi(\mu, \nu) := \left\{ \pi \in \mathcal{P}(\mathcal{X} \times \mathcal{Y}) \} \mid \int_{\mathcal{Y}} d\pi(x, y) = \mu(x), \int_{\mathcal{X}} d\pi(x, y) = \nu(y) \right\}$. When $\mu$ and $\nu$ are discrete with at most $n$ supports, the time complexity and the space complexity of the Wasserstein distance is $\mathcal{O}(n^3 \log n)$ and $\mathcal{O}(n^2)$ in turn which are very expensive. Therefore, sliced Wasserstein is proposed as an alternative solution. We first review the definition of Radon Transform.

**Radon Transform [23]** The *Radon Transform* $\mathcal{R} : \mathbb{L}_1(\mathbb{R}^d) \to \mathbb{L}_1 \left( \mathbb{R} \times \mathbb{S}^{d-1} \right)$ is defined as:

$$(\mathcal{R}f)(t, \theta) = \int_{\mathbb{R}^d} f(x)\delta(t - \langle x, \theta \rangle)dx. \tag{2}$$

Radon Transform defines a linear bijection [23]. Given a projecting direction $\theta$, $(\mathcal{R}f)(\cdot, \theta)$ is an one-dimensional function. With Radon Transform, we can now define the sliced Wasserstein distance.

**Sliced Wasserstein distance.** For $p \geq 1$, the *Sliced Wasserstein (SW)* distance [11] of $p$-th order between two distributions $\mu \in \mathcal{P}(\mathcal{X})$ and $\nu \in \mathcal{P}(\mathcal{Y})$ with an one-dimensional ground metric $c : \mathbb{R} \times \mathbb{R} \to \mathbb{R}^+$ is defined as follow:

$$\mathrm{SW}_p^p(\mu, \nu; c) = \mathbb{E}_{\theta \sim \mathcal{U}(\mathbb{S}^{d-1})}[\mathrm{W}_p^p(\mathcal{R}_\theta \sharp \mu, \mathcal{R}_\theta \sharp \nu; c)], \tag{3}$$

where $\mathcal{R}_\theta \sharp \mu$ and $\mathcal{R}_\theta \sharp \nu$ are the one-dimensional push-forward distributions created by applying Radon Transform (RT) [23] on the pdf of $\mu$ and $\nu$ with the projecting direction $\theta$. The computational benefit of SW distance comes from the closed-form solution when the one-dimensional ground metric $c(x, y) = h(x - y)$ for $h$ is a strictly convex function:

$$\mathrm{W}_p^p(\mathcal{R}_\theta \sharp \mu, \mathcal{R}_\theta \sharp \nu; c) = \int_0^1 c \left( F_{\mathcal{R}_\theta \sharp \mu}^{-1}(z), F_{\mathcal{R}_\theta \sharp \nu}^{-1}(z) \right)^p dz,$$

where $F_{\mathcal{R}_\theta \sharp \mu}^{-1}$ and $F_{\mathcal{R}_\theta \sharp \nu}^{-1}$ are inverse CDF of $\mathcal{R}_\theta \sharp \mu$ and $\mathcal{R}_\theta \sharp \nu$ respectively. When $\mu$ and $\nu$ are discrete with at most $n$ supports, the time complexity and the space complexity of the closed-form is $\mathcal{O}(n \log n)$ and $\mathcal{O}(n)$ respectively.

**Generalized Radon Transform and Generalized Sliced Wasserstein distance.** To generalize RT to non-linear operator, the *Generalized Radon Transform (GRT)* was proposed [3]. Given a defining function [26] $g : \mathbb{R}^d \times \Omega \to \mathbb{R}$, the Generalized Radon Transform [3] $\mathcal{GR} : \mathbb{L}_1(\mathbb{R}^d) \to \mathbb{L}_1 (\mathbb{R} \times \Omega)$ is defined as:

$$(\mathcal{GR}f)(t, \theta) = \int_{\mathbb{R}^d} f(x)\delta(t - g(x, \theta))dx.$$

For example, we can have the circular function [28], i.e., $g(x, \theta) = \|x - r\theta\|_2$ for $r \in \mathbb{R}^+$ and $\theta \in \Omega := \mathbb{S}^{d-1}$, homogeneous polynomials with an odd degree [51] $(m)$, i.e., $g(x, \theta) = \sum_{|\alpha|=m} \theta_\alpha x^\alpha$ with $\alpha = (\alpha_1, \ldots, \alpha_{d_\alpha}) \in \mathbb{N}^{d_\alpha}$, $|\alpha| = \sum_{i=1}^{d_\alpha} \alpha_i$, $x^\alpha = \prod_{i=1}^{d_\alpha} x_i^{\alpha_i}$, $\Omega = \mathbb{S}^{d_\alpha}$, and so on. Using GRT, the *Generalized Sliced Wasserstein (GSW)* distance is introduced in [26], which is formally defined as follow :

$$\mathrm{GSW}_p^p(\mu, \nu; c, g) = \mathbb{E}_{\theta \sim \mathcal{U}(\mathbb{S}^{d-1})}[\mathrm{W}_p^p(\mathcal{GR}_\theta^g \sharp \mu, \mathcal{GR}_\theta^g \sharp \nu; c)]. \tag{4}$$

It is worth noting that the injectivity of GRT is required to have the identity of indiscernible GSW.

**Heterogeneous joint distributions comparison.** We are given two joint distributions $\mu(x_1, x_2) \in \mathcal{P}(\mathcal{X}_1 \times \mathcal{X}_2)$ and $\nu(y_1, y_2) \in \mathcal{P}(\mathcal{Y}_1 \times \mathcal{Y}_2)$ where $X_1$ are $Y_1$ share a ground metric $c_1 : \mathcal{X}_1 \times \mathcal{Y}_1 \to \mathbb{R}^+$ and $X_2$ are $Y_2$ share a ground metric $c_2 : \mathcal{X}_2 \times \mathcal{Y}_2 \to \mathbb{R}^+$ with $(c_1 \neq c_2)$. In this case, previous works utilize the joint distribution Wasserstein distance [15, 1] to compare $\mu$ and $\nu$:

$$\mathrm{W}_p^p(\mu, \nu; c_1, c_2) := \inf_{\pi \in \Pi(\mu,\nu)} \int_{\mathcal{X}_1 \times \mathcal{X}_2 \times \mathcal{Y}_1 \times \mathcal{Y}_2} (c_1(x_1, y_1)^p + c_2(x_2, y_2)^p) d\pi(x_1, x_2, y_1, y_2), \quad (5)$$

where $\Pi(\mu, \nu) := \left\{ \pi \in \mathcal{P}(\mathcal{X}_1 \times \mathcal{X}_2 \times \mathcal{Y}_1 \times \mathcal{Y}_2) \} | \int_{\mathcal{Y}_1 \times \mathcal{Y}_2} d\pi(x_1, x_2, y_1, y_2) = \mu(x_1, x_2), \right.$

$\left. \int_{\mathcal{X}_1 \times \mathcal{X}_2} d\pi(x_1, x_2, y_1, y_2) = \nu(y_1, y_2) \right\}$. We can easily extend the definition to joint distributions with more than two marginals (see Appendix B). In contrast to the Wasserstein distance, there is no variant of SW that is designed specifically for this case. SW variants can still be used by treating $\mathcal{X}_1 \times \mathcal{X}_2$ and $\mathcal{Y}_1 \times \mathcal{Y}_2$ as homogeneous spaces $\mathcal{X}$ and $\mathcal{Y}$ which share the same Radon Transform variant and one-dimensional ground metric $c$. However, that approach cannot differentiate the difference between $\mathcal{X}_1$ and $\mathcal{X}_2$, and leverage the hierarchical structure, i.e., inside and among marginals.

## 3 Hierarchical Hybrid Sliced Wasserstein Distance

In this section, we propose the Hierarchical Hybrid Radon Transform (HHRT) which first applies P(G)RT on each marginal argument to gather each marginal information, then applies PRT on the joint transformed arguments from all marginals to gather information among marginals. After that, we introduce Hierarchical Hybrid Sliced Wasserstein distance by using HHRT as the slicing operator.

### 3.1 Hierarchical Hybrid Radon Transform

We first introduce the first building block in HHRT, i.e., Partial Generalized Radon Transform (PGRT).

**Definition 1** (Partial Generalized Radon Transform). *Given a defining function $g : \mathbb{R}^{d_1} \times \Omega \to \mathbb{R}$, Partial Generalized Radon Transform $\mathcal{PGR} : \mathbb{L}_1(\mathbb{R}^{d_1} \times \mathbb{R}^{d_2}) \to \mathbb{L}_1 \left( \mathbb{R} \times \Omega \times \mathbb{R}^{d_2} \right)$ is defined as:*

$$(\mathcal{PGR}f)(t, \theta, y) = \int_{\mathbb{R}^{d_1}} f(x, y)\delta(t - g(x, \theta))dx. \quad (6)$$

When $g(x, \theta) = \langle x, \theta \rangle$, PGRT reverts into Partial Radon Transform (PRT) [33].

**Proposition 1.** *For some defining function $g$ such as linear, circular, and homogeneous polynomials with an odd degree; the Partial Generalized Radon Transform is injective, i.e., for any functions $f_1, f_2 \in \mathbb{L}^1(\mathbb{R}^d)$, $(\mathcal{PGR}f_1)(t, \theta, y) = (\mathcal{PGR}f_2)(t, \theta, y) \, \forall t, \theta, y$ implies $f_1 = f_2$.*

The proof of Proposition 1 is given in Appendix A.1. The main idea to prove the injectivity of PGRT is to show that given a fixed $y$, the PGRT is the GRT of $f(\cdot, y)$.

**Definition 2** (Hierarchical Hybrid Radon Transform). *Given defining functions $g_1 : \mathbb{R}^{d_1} \times \Omega_1 \to \mathbb{R}$ and $g_2 : \mathbb{R}^{d_2} \times \Omega_2 \to \mathbb{R}$, Hierarchical Hybrid Radon Transform $\mathcal{HHR} : \mathbb{L}_1(\mathbb{R}^{d_1} \times \mathbb{R}^{d_2}) \to \mathbb{L}_1 \left( \mathbb{R} \times \Omega_1 \times \Omega_2 \times \mathbb{S} \right)$ is defined as:*

$$(\mathcal{HHR}f)(t, \theta_1, \theta_2, \psi) = \int_{\mathbb{R}^{d_1} \times \mathbb{R}^{d_2}} f(x_1, x_2)\delta \left( t - \psi_1 g_1(x_1, \theta_1) - \psi_2 g_2(x_2, \theta_2) \right) dx_1 dx_2, \quad (7)$$

*where $\psi = (\psi_1, \psi_2) \in \mathbb{S}$.*

The reason for using PRT for the final transform is that the previous PGRTs are assumed to be able to transform the non-linear structure to a linear line. However, PGRT can still be used as a replacement for PRT. Definition 2 can be extended to more than two marginals (see Appendix B).

**Proposition 2.** *For some defining functions $g_1, g_2$ such as linear, circular, and homogeneous polynomials with an odd degree; Hierarchical Hybrid Radon Transform is injective, i.e., for any functions $f_1, f_2 \in \mathbb{L}_1(\mathbb{R}^d)$, $(\mathcal{HHR}f_1)(t, \theta_1, \theta_2, \psi) = (\mathcal{HHR}f_2)(t, \theta_1, \theta_2, \psi) \, \forall t, \theta_1, \theta_2, \psi$ implies $f_1 = f_2$.*

The proof of Proposition 2 is given in Appendix A.2. The main idea to prove the injectivity of HHRT is to show that HHRT is the composition of PRT and multiple PGRTs.

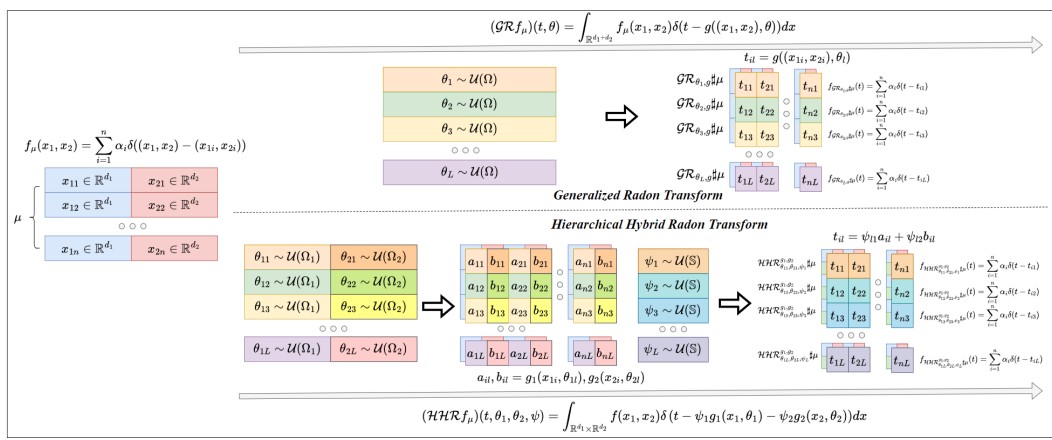

Figure 1: Generalized Radon Transform and Hierarchical Hybrid Radon Transform on a discrete distribution.

**HHRT of discrete distributions.** We are given $f(x) = \sum_{i=1}^{n} \alpha_i \delta((x_1, x_2) - (x_{1i}, x_{2i}))$ $(n \geq 1, \alpha_i \geq 0 \ \forall i)$. The HHRT of $f(x)$ is $(\mathcal{HHR}f)(t, \theta_1, \theta_2, \psi) = \sum_{i=1}^{n} \alpha_i \delta(t - \psi_1 g_1(x_{i1}, \theta_1) - \psi_2 g_2(x_{i2}, \theta_2))$. For $g_1$ and $g_2$ that are the linear function and (or) the circular function, the time complexity of the transform is $\mathcal{O}(d_1 + d_2)$ which is the same as the complexity of using RT and GRT directly. However, HHRT has an additional constant complexity scaling linearly with the number of marginals, i.e., two marginals in Definition 2.

**Example 1.** *In this paper, we focus on 3D shape data (mesh) with points and normals representation, i.e., shapes as points representation [46]. In particular, we can transform a 3D shape into a set of points and normals by sampling from the surface of the mesh. In addition, we can convert back to the 3D shape from points and normals with Poisson surface reconstruction [25] algorithm. In this setup, a shape is represented by a 6-dimensional vector $x = (x_1, x_2)$ where $x_1 \in \mathcal{X}_1 \in \mathbb{R}^3$ and $x_2 \in \mathcal{X}_2 \in \mathbb{S}^2$. For the set $\mathcal{X}_1 \in \mathbb{R}^3$, we can use directly the linear defining function $g_1(x_1, \theta_1) = \langle x_1, \theta_1 \rangle$ with $\theta_1 \in \mathbb{S}^2$. For the set $\mathcal{X}_2 \in \mathbb{S}^2$, we can utilize the circular defining function $g_2(x_2, \theta_2) = \|x_2 - r\theta_2\|_2$ with $r \in \mathbb{R}^+$ and $\theta_2 \in \mathbb{S}^2$. As alternative options for $\mathcal{X}_2$, we can use other defining functions from special cases of GRT including Vertical Slice Transform [49], Parallel Slice Transform [50], Spherical Radon Transform [6], and Stereographic Spherical Radon Transform [54].*

**Inversion.** In Proposition 2, we show that HHRT is the composition of PRT and multiple PGRTs. Therefore, the inversion of HHRT is the composition of the inversion of multiple PGRT (invertibility of PGRT depends on the choice of defining functions [3, 28]) and the inversion of PRT [23].

## 3.2 Hierarchical Hybrid Sliced Wasserstein Distance

By using HHRT, we obtain a novel variant of SW which is specifically designed for comparing heterogeneous joint distributions.

**Definitions.** We now define the Hierarchical Hybrid Sliced Wasserstein (H2SW) distance.

**Definition 3.** *For $p \geq 1$, defining functions $g_1, g_2$, the hierarchical hybrid sliced Wasserstein-p (H2SW) distance between two distributions $\mu \in \mathcal{P}(\mathcal{X}_1 \times \mathcal{X}_2)$ and $\nu \in \mathcal{P}(\mathcal{Y}_1 \times \mathcal{Y}_2)$ with an one-dimensional ground metric $c : \mathbb{R} \times \mathbb{R} \to \mathbb{R}^+$ is defined as:*

$$H2SW_p^p(\mu, \nu; c, g_1, g_2) = \mathbb{E}_{(\theta_1, \theta_2, \psi) \sim \mathcal{U}(\Omega_1 \times \Omega_2 \times \mathbb{S})}[W_p^p(\mathcal{HHR}_{\theta_1, \theta_2, \psi}^{g_1; g_2} \sharp \mu, \mathcal{HHR}_{\theta_1, \theta_2, \psi}^{g_1; g_2} \sharp \nu; c)], \quad (8)$$

*where $\mathcal{HHR}_{\theta_1, \theta_2, \psi} \sharp \mu$ and $\mathcal{HHR}_{\theta_1, \theta_2, \psi} \sharp \nu$ are the one-dimensional push-forward distributions created by applying HHRT.*

Definition 3 can be easily extended to more than two marginals (see Appendix B)

**Topological Properties.** We first show that H2SW is a valid metric on the space of distributions on any sets $\mathcal{X} \times \mathcal{Y} \in \mathbb{R}^{d_1} \times \mathbb{R}^{d_2}$ $(d_1, d_2 \geq 1)$.

**Theorem 1.** *For any $p \geq 1$, ground metric $c$, and defining functions $g_1, g_2$ which lead to the injectivity of GRT, the hierarchical hybrid sliced Wasserstein $H2SW_p(\cdot, \cdot; c, g_1, g_2)$ is a metric on $\mathcal{P}(\mathbb{R}^{d_1} \times \mathbb{R}^{d_2})$ i.e., it satisfies the symmetry, non-negativity, triangle inequality, and identity of indiscernible.*

The proof of Theorem 1 is given in Appendix A.3. It is worth noting that the identity of indiscernible property is proved by the injectivity of HHRT (Proposition 2). We now discuss the connection of H2SW to GSW and Wasserstein distance in some specific cases.

**Proposition 3.** *For any $p \geq 1$, $c(x,y) = |x - y|$, and $\mu, \nu \in \mathcal{P}(\mathbb{R}^{d_1} \times \mathbb{R}^{d_2})$, we have:*
*(i) $H2SW_p(\mu, \nu; c, g_1, g_2) \leq GSW_p(\mu_1, \nu_1; c, g_1) + GSW_p(\mu_2, \nu_2; c, g_2)$, where $\mu_1(X) = \mu(X \times \mathbb{R}^{d_2})$ and $\mu_2(Y) = \mu(\mathbb{R}^{d_1} \times Y)$ (similar with $\nu_1$ and $\nu_2$).*

*(ii) If $g_1$, $g_2$ are linear defining functions, $H2SW_p(\mu, \nu; c, g_1, g_2) \leq W_p(\mu_1, \nu_1; c) + W_p(\mu_2, \nu_2; c)$.*

*(iii) If $p = 1$, $g_1$, $g_2$ are linear defining functions, $H2SW_1(\mu, \nu; c, g_1, g_2) \leq W_1(\mu, \nu; c)$.*

The proof of Proposition 3 is given in Appendix A.4.

**Sample Complexity.** We now discuss the sample complexity of H2SW.

**Proposition 4.** *For any $p \geq 1$, dimension $d_1, d_2 \geq 1$, $q > p$, $c(x,y) = |x - y|$, $g_1, g_2$ are linear defining functions or circular defining functions, and $\mu, \nu \in \mathcal{P}_q(\mathbb{R}^{d_1} \times \mathbb{R}^{d_2})$ with the corresponding empirical distributions $\mu_n$ and $\nu_n$ ($n \geq 1$), there exists a constant $C_{p,q}$ depending on $p, q$ such that:*

$$\mathbb{E} \left| H2SW_p(\mu_n, \nu_n; c, g_1, g_2) - H2SW_p(\mu, \nu; c, g_1, g_2) \right|$$

$$\leq C_{p,q}^{\frac{1}{p}} \left( \sum_{i=0}^{q} q^i C_{g_1,g_2}^{q-i} (M_i(\mu) + M_i(\nu)) \right)^{\frac{1}{p}} \begin{cases} n^{-1/2p} \text{ if } q > 2p, \\ n^{-1/2p} \log(n)^{\frac{1}{p}} \text{ if } q = 2p, \\ n^{-(q-p)/pq} \text{ if } q \in (p, 2p), \end{cases} \quad (9)$$

*where $M_q(\mu)$ and $M_q(\nu)$ are the $q$-th moments of $\mu$ and $\nu$, $C_{g_1,g_2}$ is a constant depends on $g_1$, $g_2$.*

The proof of Proposition 4 is given in Appendix A.5. The rate in Proposition 4 is as good as the rate of SW in [38], however, it is slightly worse than the rate of SW in [44, 36, 5] due to the usage of the circular defining functions and simpler assumptions. To the best of our knowledge, the sample complexity of GSW has not been investigated.

**Monte Carlo Estimation.** Since the expectation in H2SW (Equation 8) is intractable, Monte Carlo estimation and Quasi-Monte Carlo approximation [39] can be used to form a practical evaluation of H2SW. Here, we utilize Monte Carlo estimation for simplicity. In particular, we sample $\theta_{11}, \ldots, \theta_{1L} \overset{i.i.d}{\sim} \mathcal{U}(\Omega_1)$, $\theta_{21}, \ldots, \theta_{2L} \overset{i.i.d}{\sim} \mathcal{U}(\Omega_2)$, and $\psi_1, \ldots, \psi_L \overset{i.i.d}{\sim} \mathcal{U}(\mathbb{S})$. After that, we form the following estimation of H2SW:

$$\widehat{H2SW}_p^p(\mu, \nu; c, g_1, g_2, L) = \frac{1}{L} \sum_{l=1}^{L} W_p^p(\mathcal{HHR}_{\theta_{1l},\theta_{2l},\psi_l}^{g_1,g_2} \sharp \mu, \mathcal{HHR}_{\theta_{1l},\theta_{2l},\psi_l}^{g_1,g_2} \sharp \nu; c). \quad (10)$$

**Proposition 5.** *For any $p \geq 1$, dimension $d_1, d_2 \geq 1$, and $\mu, \nu \in \mathcal{P}(\mathbb{R}_1^d \times \mathbb{R}^{d_2})$, we have:*

$$\mathbb{E} | \widehat{H2SW}_p^p(\mu, \nu; c, g_1, g_2, L) - H2SW_p^p(\mu, \nu; c, g_1, g_2)|$$

$$\leq \frac{1}{\sqrt{L}} Var \left[ W_p^p(\mathcal{HHR}_{\theta_1,\theta_2,\psi}^{g_1,g_2} \sharp \mu, \mathcal{HHR}_{\theta_1,\theta_2,\psi}^{g_1,g_2} \sharp \nu; c) \right]^{\frac{1}{2}}, \quad (11)$$

*where the variance is with respect to $\mathcal{U}(\Omega_1 \times \Omega_2 \times \mathbb{S})$.*

The proof of Proposition 5 is given in Appendix A.6. From the proposition, we see that the estimation error of H2SW is the same as SW which is $\mathcal{O}(L^{-1/2})$.

**Computational Complexities.** The time complexity and memory complexity of H2SW with linear and circular defining functions are $\mathcal{O}(Ln \log n + L(d_1 + d_2 + k)n)$ and $\mathcal{O}(Ln + (d_1 + d_2 + k)n)$ with $k$ is the number of marginals i.e., 2. We can see that the complexities of H2SW are the same as those of SW in terms of the number of supports $n$ and the number of dimensions $d$. We demonstrate the process of HHRT compared to GRT on a discrete distribution with $L$ realization of $\theta_1, \theta_2, \psi$ in Figure 1. Overall, the complexities of defining functions are often different in the number of dimensions, hence, H2SW is always scaled the same as SW in the number of supports i.e., $\mathcal{O}(n \log n)$.

**Gradient Estimation.** In applications, it is desirable to estimate the gradient $\nabla_\phi H2SW_p^p(\mu_\phi, \nu; c, g_1, g_2)$. We can move the gradient operator to inside the expectation

Table 1: Summary of joint Wasserstein distances across time steps from deformation from the sphere mesh to the Armadillo mesh.

| Distances | Step 100 ($W_{c_1,c_2}$ ↓) | Step 300 ($W_{c_1,c_2}$ ↓) | Step 500 ($W_{c_1,c_2}$ ↓) | Step 1500 ($W_{c_1,c_2}$ ↓) | Step 4000 ($W_{c_1,c_2}$ ↓) | Step 5000 ($W_{c_1,c_2}$ ↓) |
|---|---|---|---|---|---|---|
| SW L=10 | 1852.519±3.236 | 1436.686±3.056 | 1071.227±2.449 | 104.452±2.35 | **6.19±0.307** | 2.726±0.305 |
| GSW L=10 | 1893.438±3.205 | 1535.737±3.363 | 1192.52±3.274 | 143.518±1.04 | 8.73±0.353 | 4.743±0.134 |
| H2SW L=10 | **1840.73±1.282** | **1422.667±7.813** | **1058.171±5.362** | **95.672±4.376** | 6.326±0.151 | **2.602±0.201** |
| SW L=100 | 1847.572±0.303 | 1426.425±0.528 | 1059.127±1.106 | 89.693±0.793 | **4.453±0.22** | 1.171±0.056 |
| GSW L=100 | 1889.312±0.883 | 1525.269±1.078 | 1179.1±2.052 | 122.618±1.175 | 7.905±0.373 | 3.226±0.388 |
| H2SW L=100 | **1839.347±1.986** | **1417.1±3.677** | **1048.895±4.008** | **86.078±0.623** | 4.61±0.431 | **1.086±0.177** |

Table 2: Summary of joint Wasserstein distances (multiplied by 100) across time steps from deformation from the sphere mesh to the Stanford Bunny mesh.

| Distances | Step 100 ($W_{c_1,c_2}$ ↓) | Step 300 ($W_{c_1,c_2}$ ↓) | Step 500 ($W_{c_1,c_2}$ ↓) | Step 1500 ($W_{c_1,c_2}$ ↓) | Step 4000 ($W_{c_1,c_2}$ ↓) | Step 5000 ($W_{c_1,c_2}$ ↓) |
|---|---|---|---|---|---|---|
| SW L=10 | 26.868±0.579 | 4.46±0.195 | 1.52±0.081 | **0.623±0.024** | 0.221±0.023 | 0.14±0.018 |
| GSW L=10 | 26.837±0.496 | 4.378±0.128 | 1.548±0.062 | 0.653±0.01 | **0.173±0.018** | 0.146±0.013 |
| H2SW L=10 | **23.283±0.119** | **2.221±0.124** | **1.452±0.075** | 0.636±0.045 | 0.177±0.009 | **0.089±0.022** |
| SW L=100 | 26.678±0.168 | 4.109±0.138 | 1.458±0.142 | **0.362±0.023** | 0.072±0.017 | 0.049±0.006 |
| GSW L=100 | 26.795±0.202 | 4.084±0.109 | 1.375±0.049 | 0.372±0.026 | **0.048±0.004** | 0.042±0.017 |
| H2SW L=100 | **23.772±0.19** | **2.388±0.009** | **1.358±0.051** | 0.488±0.026 | 0.064±0.01 | **0.038±0.007** |

and then apply Monte Carlo estimation. The gradient $\nabla_\phi W_p^p(\mathcal{H}\mathcal{H}\mathcal{R}_{\theta_1,\theta_2,\psi}^{g_1;g_2}\sharp\mu_\phi, \mathcal{H}\mathcal{H}\mathcal{R}_{\theta_1,\theta_2,\psi}^{g_1;g_2}\sharp\nu; c)]$ can be computed easily since the functions $g_1, g_2$ are usually differentiable.

**Beyond uniform slicing distribution.** H2SW is defined with the uniform slicing distribution in Definition 3, however, it is possible to extend it to other slicing distributions such as the maximal projecting direction [18], distributional slicing distribution [42], and energy-based slicing distribution [41]. Since the choice of slicing distribution is independent of the main contribution i.e., the slicing operator, we leave this investigation to future work.

**H2SW for distributions on the product of Hadamard manifolds.** A recent work [9] extends sliced Wasserstein on hyperbolic manifolds [7] and on the manifold of symmetric positive definite matrices [10] to Hadamard manifolds i.e., manifold non-positive curvature. The work discusses the extension of SW to the product of Hadamard manifolds. For the geodesic projection, the closed-form for the projection is intractable. For the Busemann projection, the Busemann projection on the product manifolds is the weighted sum of the Busemann projection with the weights belonging to the unit-sphere. In the work, the weights are a fixed hyperparameter i.e., Cartan-Hadamard Sliced-Wasserstein (CHSW) utilizes only one Busemann function to project the joint distribution. In contrast, H2SW utilizes the Radon Transform on the joint spaces of projections i.e., considering all distributed weighted combinations which is equivalent to considering all Busemann functions under a probability law. As a result, the H2SW is a valid metric as long as the Busemann projections can be proven to be injective (the injectivity of the Busemann projection has not been known at the moment) while Cartan-Hadamard Sliced-Wasserstein is only pseudo metric since the injectivity of a fixed weighted combination is not trivial to show. Moreover, H2SW does not only focus on the product of Hadamard manifolds i.e., H2SW is a generic distance for heterogeneous joint distributions in which marginal domains are not necessary manifolds e.g., images [40], functions [21], and so on. In the later experiments, we conduct experiments on comparing 3D shapes which are represented by a distribution on the product of the Euclidean space and the 2D sphere (not a Hadamard manifold).

## 4 Experiments

In this section, we first compare the performance of the proposed H2SW with SW and GSW in the 3D mesh deformation application. After that, we further evaluate the performance of H2SW in training a deep 3D mesh autoencoder compared to SW and GSW. Finally, we compare H2SW with SW and Cartan-Hadamard Sliced-Wasserstein (CHSW) in datasets comparison on the product of Hadamard manifolds. In the experiments, we use $c(x,y) = |x-y|$ and $p = 2$ for all SW variants.

### 4.1 3D Mesh Deformation

In this task, we would like to move from a source mesh to a target mesh. To represent those meshes, we sample 10000 points by Poisson disk sampling and their corresponding normal vectors of the mesh surface at those points. Let the source mesh be denoted as $X(0) = \{x_1(0), \ldots, x_n(0)\}$ and the target mesh be denoted as $Y = \{y_1, \ldots, y_n\}$. We deform $X(0)$ to $Y$ by integrating the ordinary differential equation $\dot{X}(t) = -n\nabla_{X(t)}\left[\mathcal{S}\left(\frac{1}{n}\sum_{i=1}^n \delta(x-x_i(t)), \frac{1}{n}\sum_{i=1}^n \delta(y-y_i)\right)\right]$, where $\mathcal{S}$ denotes a SW

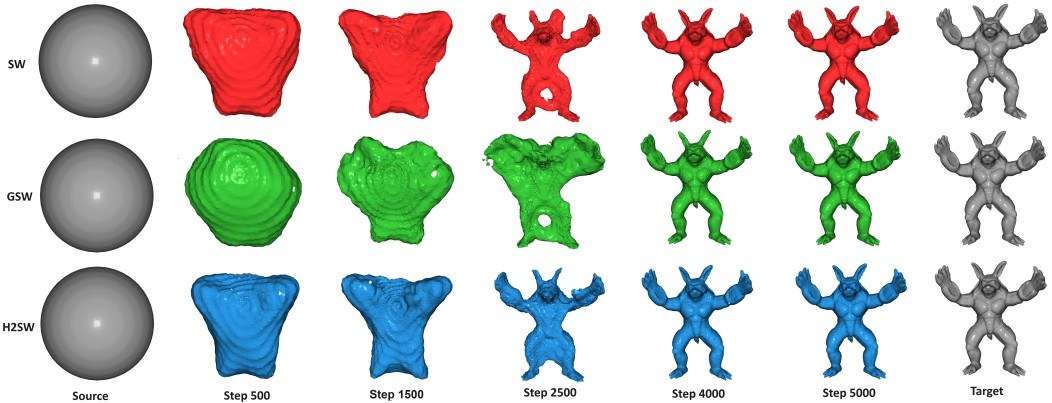

Figure 2: Visualization of deformation from the sphere mesh to the Armadillo mesh with $L = 10$.

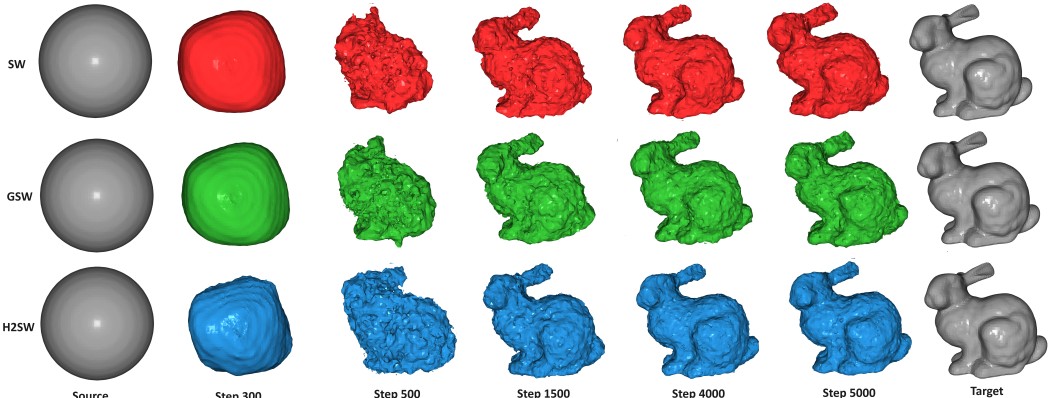

Figure 3: Visualization of deformation from the sphere mesh to the Stanford Bunny mesh with $L = 10$.

variant. We utilize the Euler discretization scheme with step size 0.01 and 5000 steps. The normal vectors are projected back to the sphere after taking an Euler step. For evaluation, we use the joint Wasserstein distance in Equation 5 with the mixed distance from the Euclidean distance and the great circle distance. We use the circular defining function for GSW, and use the linear defining function and the circular defining function for H2SW. We vary the number of projections $L \in \{10, 100\}$ for all variants. For H2SW and GSW, we select the best hyperparameter of the circular defining function $r \in \{0.5, 0.7, 0.8, 0.9, 1, 5, 10, 50, 100\}$.

**Results.** We compare H2SW with GSW and SW by deforming the sphere mesh to the Armadillo mesh [59]. We report the quantitative results in Table 1 after 3 independent runs and the qualitative result for $L = 10$ in Figure 2 and $L = 100$ in Figure 6 in Appendix D. From Table 1, we observe that H2SW helps the deformation convergence faster at the beginning and better at the end in terms of the joint Wasserstein distance, especially for a small value of the number of projections i.e., $L = 10$. The result for $L = 100$ is better than $L = 10$ which is consistent with Proposition 5. The qualitative results in Figure 2 and Figure 6 also reinforce the favorable performance of H2SW since they are visually consistent with quantitative scores. We also conduct deformation to the Stanford Bunny mesh [16, 59] in Table 2, Figure 3, and Figure 7 in Appendix D and we observe the same phenomenon that H2SW is the best variant for 3D meshes. From those experiments, H2SW has shown the benefit of the HHRT in transforming a joint distribution over the product of the Euclidean space and the 2D sphere compared to the conventional RT of SW and the conventional GRT of GSW.

## 4.2 Training deep 3D mesh autoencoder

We utilize the processed ShapeNet dataset [12] from [46], then sample 2048 points and the corresponding normal vectors from each shape in the dataset. Formally, we would like to train an autoencoder that contains an encoder $f_\phi$ that maps a mesh $X \in \mathbb{R}^{2048 \times 6}$ to a latent code $z \in \mathbb{R}^{1024}$, and a decoder $g_\psi$ that maps the latent code $z$ back to the reconstructed mesh $\tilde{X} \in \mathbb{R}^{2048 \times 6}$. We adopt

Table 3: Joint Wasserstein distance reconstruction errors (multiplied by 100) from three different runs of autoencoders trained by SW, GSW, and H2SW with the number of projections $L = 100$ and $L = 1000$.

| Distance | Epoch 500 $W_{c_1,c_2}(\downarrow)$ | Epoch 1000 $W_{c_1,c_2}(\downarrow)$ | Epoch 2000 $W_{c_1,c_2}(\downarrow)$ |
|---|---|---|---|
| SW L=100 | $136.87 \pm 1.18$ | $133.24 \pm 0.50$ | $131.10 \pm 0.34$ |
| GSW L=100 | $136.51 \pm 0.20$ | $133.36 \pm 0.24$ | $130.80 \pm 0.46$ |
| H2SW L=100 | $\mathbf{135.54 \pm 0.72}$ | $\mathbf{132.24 \pm 0.36}$ | $\mathbf{130.24 \pm 0.47}$ |
| SW L=1000 | $135.85 \pm 0.92$ | $132.88 \pm 0.27$ | $130.93 \pm 0.11$ |
| GSW L=1000 | $136.40 \pm 0.10$ | $133.02 \pm 0.98$ | $130.76 \pm 0.26$ |
| H2SW L=1000 | $\mathbf{135.47 \pm 0.64}$ | $\mathbf{132.17 \pm 0.10}$ | $\mathbf{129.87 \pm 0.44}$ |

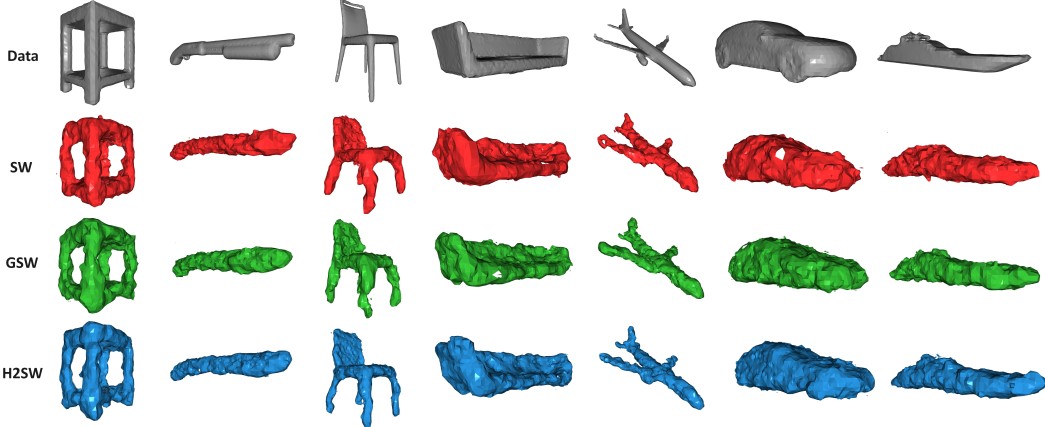

Figure 4: Visualization of some randomly selected reconstruction meshes from autoencoders trained by SW, GSW, and H2SW in turn with the number of projections $L = 100$ at epoch 2000.

Point-Net [48] architecture to construct the autoencoder. We want to train the encoder $f_\phi$ and the decoder $g_\psi$ such that $\tilde{X} = g_\psi(f_\phi(X)) \approx X$ for all shapes $X$ in the dataset. To do that, we solve the following optimization problem:

$$\min_{\phi,\gamma} \mathbb{E}_{X \sim \mu(X)}[\mathcal{S}(P_X, P_{g_\gamma(f_\phi(X))})],$$

where $\mathcal{S}$ is a sliced Wasserstein variant, and $P_X = \frac{1}{n} \sum_{i=1}^{n} \delta(x - x_i)$ denotes the empirical distribution over the point cloud $X = (x_1, \ldots, x_n)$. We train the autoencoder for 2000 epochs on the training set of the ShapeNet dataset using an SGD optimizer with a learning rate of $1e-3$, and a batch size of 128. For evaluation, we also use the joint Wasserstein distance in Equation 5 with the mixed distance from the Euclidean distance and the great circle distance to measure the average reconstruction loss on the testing set of the ShapeNet dataset. We use the circular defining function for GSW, and use the linear defining function and the circular defining function for H2SW. For H2SW and GSW, we select the best hyperparameter of the circular defining function $r \in \{0.5, 0.7, 0.8, 0.9, 1, 5, 10\}$. For more details such as the neural network architectures, we refer the reader to Appendix B.

**Results.** We report the joint Wasserstein reconstruction errors (measured in three independent times) on the testing set in Table 3 with trained autoencoder at epoch 500, 1000, and 2000 from SW, GSW, and H2SW with the number of projections $L = 100$ and $L = 1000$. In addition, we show some randomly reconstructed meshes for epoch 2000in Figure 4 and for epoch 500 in Figure 8 in Appendix D. From Table 3, we observe that H2SW yields the lowest reconstruction errors for both $L = 100$ and $L = 1000$. Moreover, we see that the reconstruction errors are lower with $L = 1000$ than ones with $L = 100$ for all SW variants. The qualitative reconstructed meshes in Figure 4 and Figure 8reflect the same relative comparison. It is worth noting that both the qualitative and the qualitative performance of autoencoders can be improved by using more powerful neural networks. Since we focus on comparing SW, GSW, and H2SW, we only use a light neural network i.e., Point-Net [48] architecture. The trained autoencoders can be further used to reduce the size of 3D meshes for data compression and for dimension reduction, however, such downstream applications are not our focus in the current investigation of the paper.

Table 4: Relative error to the joint Wasserstein distance of SW, CHSW, and H2SW.

| Distances | $L = 100$ | $L = 500$ | $L = 1000$ | $L = 2000$ |
|---|---|---|---|---|
| SW | $4.618 \pm 0.744$ | $4.253 \pm 0.398$ | $4.235 \pm 0.310$ | $4.198 \pm 0.238$ |
| CHSW | $4.449 \pm 0.497$ | $4.063 \pm 0.254$ | $4.059 \pm 0.167$ | $4.035 \pm 0.145$ |
| H2SW | $\mathbf{4.381 \pm 0.695}$ | $\mathbf{4.001 \pm 0.267}$ | $\mathbf{4.048 \pm 0.182}$ | $\mathbf{3.998 \pm 0.142}$ |

Figure 5: Cost matrices between datasets from SW, CHSW, and H2SW with $L = 2000$.

## 4.3 Comparing Datasets on The Product of Hadamard Manifolds

We follow the same experimental setting from [9]. Here, we have datasets as sets of feature-label pairs which are embedded in the space of $\mathbb{R}^{d_1} \times \mathbb{L}^{d_2}$ where $\mathbb{L}^{d_2}$ denotes a Lorentz model of $d_2$ dimension (a hyperbolic space). We uses MNIST [31] dataset, EMNIST dataset [14], Fashion MNIST dataset [56], KMNIST dataset [13], and USPS dataset [24]. For CHSW, we use Busemann projection on the product space of Euclidean and the Lorentz model. For H2SW, we use the linear defining function and the Busemann function on the Lorentz model. We refer the reader to Appendix B for greater detail on Busemann functions and experimental setups. We compare SW, CHSW, and H2SW by varying $L \in \{100, 500, 1000, 2000\}$. For evaluation, we use the joint Wasserstein distance in [1] as the ground truth. In particular, let $C_W$ be the cost matrix from the joint Wasserstein distance and $C$ be a given cost matrix, we use $|C/\max(C) - C_W/\max(C_W)|$ as the relative error.

**Results.** We report the relative errors from SW, CHSW, and H2SW in Table 4 after 100 independent runs. In addition, we show the cost matrices from SW, CHSW, H2SW. and joint Wasserstein distance with $L = 2000$ in Figure 5. Cost matrices for $L = 100$, $L = 500$, and $L = 1000$ are given in Figure 9- 11 in Appendix D. From Table 4, we see that H2SW gives a lower relative error than CHSW and SW. Therefore, using H2SW for comparing datasets is the most equivalent to the joint Wasserstein distance in terms of the relative error. We also observe that increasing the value of the number of projections also reduces the relative errors for all SW variants. Again, we would like to recall that H2SW can be used for heterogeneous joint distributions beyond the product of Hadamard manifolds as shown in previous experiments.

## 5 Conclusion

We have presented Hierarchical Hybrid Sliced Wasserstein (H2SW) distance, a novel sliced probability metric for heterogeneous joint distributions i.e., joint distributions have marginals on different domains. The key component of H2SW is the proposed hierarchical hybrid Radon Transform (HHRT) which is the composition of partial Radon Transform and multiples proposed partial generalized Radon Transform. We then discuss the injectivity of the proposed transforms and theoretical properties of H2SW including topological properties, statistical properties, and computational properties. On the experimental side, we show that H2SW has favorable performance in applications of 3D mesh deformation, training deep 3D mesh autoencoder, and datasets comparison. In those applications, heterogeneous joint distributions appear in the form of joint distributions on the product of Euclidean space and 2D sphere, and the product of Hadamard manifolds. In the future, we will extend the application of H2SW to more complicated heterogeneous joint distributions.

## Acknowledgements

NH acknowledges support from the NSF IFML 2019844 and the NSF AI Institute for Foundations of Machine Learning.

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

# Supplement to "Hierarchical Hybrid Sliced Wasserstein: A Scalable Metric for Heterogeneous Joint Distributions"

We first provide skipped proofs in the main paper in Appendix A. We then provide some additional materials including additional background and extended definitions in Appendix B. After that, we discuss some related works in Appendix C. We report additional experimental results in Appendix D. Finally, we report computational infrastructure in Appendix E.

## A    Proofs

### A.1    Proof of Proposition 1

For any $t, \theta, y$, we are given $(\mathcal{PGR}f_1)(t, \theta, y) = (\mathcal{PGR}f_2)(t, \theta, y)$. By Definition 1, we have:

$$\int_{\mathbb{R}^{d_1}} f_1(x, y)\delta(t - g(x, \theta))dx = \int_{\mathbb{R}^{d_1}} f_2(x, y)\delta(t - g(x, \theta))dx.$$

For any $\varepsilon \in \mathbb{R}^{d_2}$, we have:

$$\int_{\mathbb{R}^{d_2}} \int_{\mathbb{R}^{d_1}} f_1(x, y)\delta(t - g(x, \theta))e^{-i2\pi\langle\varepsilon, y\rangle}dxdy = \int_{\mathbb{R}^{d_2}} \int_{\mathbb{R}^{d_1}} f_2(x, y)\delta(t - g(x, \theta))e^{-i2\pi\langle\varepsilon, y\rangle}dxdy.$$

Applying the Fubini's theorem, we have:

$$\int_{\mathbb{R}^{d_1}} f_1(x, y) \int_{\mathbb{R}^{d_2}} e^{-i2\pi\langle\varepsilon, y\rangle}dy\delta(t - g(x, \theta))dx = \int_{\mathbb{R}^{d_1}} \int_{\mathbb{R}^{d_2}} f_2(x, y)e^{-i2\pi\langle\varepsilon, y\rangle}dy\delta(t - g(x, \theta))dx,$$

which is:

$$\left(\mathcal{GR} \int_{\mathbb{R}^{d_2}} f_1(x, y)e^{-i2\pi\langle\varepsilon, y\rangle}dy\right) = \left(\mathcal{GR} \int_{\mathbb{R}^{d_2}} f_2(x, y)e^{-i2\pi\langle\varepsilon, y\rangle}dy\right).$$

By the injectivity of GRT, we have:

$$\int_{\mathbb{R}^{d_2}} f_1(x, y)e^{-i2\pi\langle\varepsilon, y\rangle}dy = \int_{\mathbb{R}^{d_2}} f_2(x, y)e^{-i2\pi\langle\varepsilon, y\rangle}dy.$$

Then, for any $\epsilon \in \mathbb{R}^{d_1}$, we have

$$\int_{\mathbb{R}^{d_1}} \int_{\mathbb{R}^{d_2}} f_1(x, y)e^{-i2\pi\langle\varepsilon, y\rangle}e^{-i2\pi\langle\epsilon, x\rangle}dydx = \int_{\mathbb{R}^{d_1}} \int_{\mathbb{R}^{d_2}} f_2(x, y)e^{-i2\pi\langle\varepsilon, y\rangle}e^{-i2\pi\langle\epsilon, x\rangle}dydx.$$

which is $(\mathcal{F}f_1(x, y)) = (\mathcal{F}f_2(x, y)))$ with $\mathcal{F}$ denotes the Fourier transform. By the injectivity of the Fourier Transform, we have $f_1(x, y) = f_2(x, y)$ for any $x, y$, which concludes the proof.

### A.2    Proof of Proposition 2

We first show that HHRT is the composition of PGRT and PRT. We have

$$(\mathcal{PR}(\mathcal{PGR}(\mathcal{PGR}f)))(t, \theta_1, \theta_2, \psi)$$
$$= \int_{\mathbb{R}^2} \int_{\mathbb{R}^{d_1}} \int_{\mathbb{R}^{d_2}} f(x, y)\delta(t_1 - g_1(x, \theta_1))\delta(t_2 - g_2(y, \theta_2))\delta(t - \psi_1 t_1 - \psi_2 t_2)dxdydt_1dt_2$$
$$= \int_{\mathbb{R}^{d_1}} \int_{\mathbb{R}^{d_2}} f(x, y) \int_{\mathbb{R}^2} \delta(t_1 - g_1(x, \theta_1))\delta(t_2 - g_2(y, \theta_2))\delta(t - \psi_1 t_1 - \psi_2 t_2)dt_1dt_2dxdy$$
$$= \int_{\mathbb{R}^{d_1}} \int_{\mathbb{R}^{d_2}} f(x, y)\delta\left(t - \psi_1 g_1(x, \theta_1) - \psi_2 g_2(y, \theta_2)\right)dxdy$$
$$= (\mathcal{HHR}f)(t, \theta_1, \theta_2, \psi).$$

For any $t, \theta_1, \theta_2, \psi$, we are given $(\mathcal{HHR}f_1)(t, \theta_1, \theta_2, \psi) = (\mathcal{HHR}f_2)(t, \theta_1, \theta_2, \psi)$, which is equivalent to:

$$(\mathcal{PR}(\mathcal{PGR}(\mathcal{PGR}f_1)))(t, \theta_1, \theta_2, \psi) = (\mathcal{PR}(\mathcal{PGR}(\mathcal{PGR}f_2)))(t, \theta_1, \theta_2, \psi).$$

By the injectivity of the PRT and the PGRT, we obtain $f_1(x, y) = f_2(x, y)$ for any $x, y$ which completes the proof.

## A.3 Proof of Theorem 1

To prove that the hierarchical hybrid sliced Wasserstein $H2SW_p(\cdot, \cdot; c, g_1, g_2)$ is a metric on the space of distributions on $\mathcal{P}(\mathbb{R}^{d_1} \times \mathbb{R}^{d_2})$ for any $p \geq 1$, ground metric $c$, and defining functions $g_1, g_2$, we need to show that it satisfies non-negativity, symmetry, triangle inequality, and identity of indiscernible.

**Non-Negativity.** Since $\mathrm{W}_p^p(\mathcal{HHR}_{\theta_1,\theta_2,\psi}^{g_1;g_2}\sharp\mu, \mathcal{HHR}_{\theta_1,\theta_2,\psi}^{g_1;g_2}\sharp\nu; c) \geq 0$ [47] for any $\theta_1, \theta_2, \psi$, we have:

$$\mathbb{E}_{(\theta_1,\theta_2,\psi)\sim\mathcal{U}(\Omega_1\times\Omega_2\times\mathbb{S})}[\mathrm{W}_p^p(\mathcal{HHR}_{\theta_1,\theta_2,\psi}^{g_1;g_2}\sharp\mu, \mathcal{HHR}_{\theta_1,\theta_2,\psi}^{g_1;g_2}\sharp\nu; c)] \geq 0,$$

which means that $H2SW_p(\mu, \nu; c, g_1, g_2) \geq 0$ for any $\mu$ and $\nu$.

**Symmetry.** Since we have the symmetry of the Wasserstein distance $\mathrm{W}_p^p(\mathcal{HHR}_{\theta_1,\theta_2,\psi}^{g_1;g_2}\sharp\mu, \mathcal{HHR}_{\theta_1,\theta_2,\psi}^{g_1;g_2}\sharp\nu; c) = \mathrm{W}_p^p(\mathcal{HHR}_{\theta_1,\theta_2,\psi}^{g_1;g_2}\sharp\nu, \mathcal{HHR}_{\theta_1,\theta_2,\psi}^{g_1;g_2}\sharp\mu; c)$ [47] for any $\theta_1, \theta_2, \psi$, we have:

$$\mathbb{E}_{(\theta_1,\theta_2,\psi)\sim\mathcal{U}(\Omega_1\times\Omega_2\times\mathbb{S})}[\mathrm{W}_p^p(\mathcal{HHR}_{\theta_1,\theta_2,\psi}^{g_1;g_2}\sharp\mu, \mathcal{HHR}_{\theta_1,\theta_2,\psi}^{g_1;g_2}\sharp\nu; c)]$$
$$= \mathbb{E}_{(\theta_1,\theta_2,\psi)\sim\mathcal{U}(\Omega_1\times\Omega_2\times\mathbb{S})}[\mathrm{W}_p^p(\mathcal{HHR}_{\theta_1,\theta_2,\psi}^{g_1;g_2}\sharp\nu, \mathcal{HHR}_{\theta_1,\theta_2,\psi}^{g_1;g_2}\sharp\mu; c)],$$

which means that $H2SW_p(\mu, \nu; c, g_1, g_2) = H2SW_p(\nu, \mu; c, g_1, g_2)$ any $\mu$ and $\nu$.

**Triangle Inequality.** Given $c$ to be a valid metric on $\mathbb{R}$, we can use the triangle inequality of the Wasserstein distance. For any distributions $\mu_1, \mu_2, \mu_3 \in \mathcal{P}(\mathbb{R}^{d_1} \times \mathbb{R}^{d_2})$, we have:

$$H2SW_p(\mu_1, \mu_2; c, g_1, g_2) = \left(\mathbb{E}_{(\theta_1,\theta_2,\psi)\sim\mathcal{U}(\Omega_1\times\Omega_2\times\mathbb{S})}[\mathrm{W}_p^p(\mathcal{HHR}_{\theta_1,\theta_2,\psi}^{g_1;g_2}\sharp\mu_1, \mathcal{HHR}_{\theta_1,\theta_2,\psi}^{g_1;g_2}\sharp\mu_2; c)]\right)^{\frac{1}{p}}$$

$$\leq \left(\mathbb{E}_{(\theta_1,\theta_2,\psi)\sim\mathcal{U}(\Omega_1\times\Omega_2\times\mathbb{S})}[(\mathrm{W}_p(\mathcal{HHR}_{\theta_1,\theta_2,\psi}^{g_1;g_2}\sharp\mu_1, \mathcal{HHR}_{\theta_1,\theta_2,\psi}^{g_1;g_2}\sharp\mu_3; c)\right.$$
$$\left. +\mathrm{W}_p(\mathcal{HHR}_{\theta_1,\theta_2,\psi}^{g_1;g_2}\sharp\mu_3, \mathcal{HHR}_{\theta_1,\theta_2,\psi}^{g_1;g_2}\sharp\mu_2; c))^p]\right)^{\frac{1}{p}}$$

$$\leq \left(\mathbb{E}_{(\theta_1,\theta_2,\psi)\sim\mathcal{U}(\Omega_1\times\Omega_2\times\mathbb{S})}[\mathrm{W}_p^p(\mathcal{HHR}_{\theta_1,\theta_2,\psi}^{g_1;g_2}\sharp\mu_1, \mathcal{HHR}_{\theta_1,\theta_2,\psi}^{g_1;g_2}\sharp\mu_3; c)]\right)^{\frac{1}{p}}$$
$$+ \left(\mathbb{E}_{(\theta_1,\theta_2,\psi)\sim\mathcal{U}(\Omega_1\times\Omega_2\times\mathbb{S})}[\mathrm{W}_p^p(\mathcal{HHR}_{\theta_1,\theta_2,\psi}^{g_1;g_2}\sharp\mu_3, \mathcal{HHR}_{\theta_1,\theta_2,\psi}^{g_1;g_2}\sharp\mu_2; c)]\right)^{\frac{1}{p}}$$
$$= H2SW_p(\mu_1, \mu_3; c, g_1, g_2) + H2SW_p(\mu_3, \mu_2; c, g_1, g_2),$$

where the final inequality is due to Minkowski's inequality. Therefore, we complete the proof for the triangle inequality of the hierarchical hybrid sliced Wasserstein.

**Identity of indiscernible.** For any $p \geq 1$, ground metric $c$, and $g_1, g_2$, when $\mu = \nu$, we have $\mathcal{HHR}_{\theta_1,\theta_2,\psi}^{g_1;g_2}\sharp\mu = (\mathcal{HHR}_{\theta_1,\theta_2,\psi}^{g_1;g_2}\sharp\nu$. Therefore, we have $\mathrm{W}_p^p(\mathcal{HHR}_{\theta_1,\theta_2,\psi}^{g_1;g_2}\sharp\mu_1, \mathcal{HHR}_{\theta_1,\theta_2,\psi}^{g_1;g_2}\sharp\mu_2; c) = 0$ which leads to $H2SW_p(\mu, \nu; c, g_1, g_2) = 0$. Now, assume that $H2SW_p(\mu, \nu; c, g_1, g_2) = 0$, then $\mathrm{W}_p^p(\mathcal{HHR}_{\theta_1,\theta_2,\psi}^{g_1;g_2}\sharp\mu_1, \mathcal{HHR}_{\theta_1,\theta_2,\psi}^{g_1;g_2}\sharp\mu_2; c) = 0$ for almost everywhere $\theta_1 \in \Omega_1, \theta_2 \in \Omega_2, \psi \in \mathbb{S}$. By applying the identity property of the Wasserstein distance, we have $\mathcal{HHR}_{\theta_1,\theta_2,\psi}^{g_1;g_2}\sharp\mu = (\mathcal{HHR}_{\theta_1,\theta_2,\psi}^{g_1;g_2}\sharp\nu$ for almost everywhere $\theta_1 \in \Omega_1, \theta_2 \in \Omega_2, \psi \in \mathbb{S}$. Since the HHRT is injective (proved in Proposition 2), we obtain $\mu = \nu$.

## A.4 Proof of Proposition 3

(i) For any $p \geq 1$, $c(x, y) = |x - y|$, and $\mu, \nu \in \mathcal{P}(\mathbb{R}^{d_1} \times \mathbb{R}^{d_2})$, we have:

$H2SW_p(\mu, \nu; c, g_1, g_2)$

$$= \left(\mathbb{E}_{(\theta_1,\theta_2,\psi)\sim\mathcal{U}(\Omega_1\times\Omega_2\times\mathbb{S})}[\mathrm{W}_p^p(\mathcal{HHR}_{\theta_1,\theta_2,\psi}^{g_1;g_2}\sharp\mu, \mathcal{HHR}_{\theta_1,\theta_2,\psi}^{g_1;g_2}\sharp\nu; c)]\right)^{\frac{1}{p}}$$

$$= \left(\mathbb{E}\left[\inf_{\pi\in\Pi(\mu,\nu)}\int |\psi_1(g_1(\theta_1, x_1) - g_1(\theta_1, y_1)) + \psi_2(g_2(\theta_2, x_2) - g_1(\theta_2, y_2))|^p d\pi(x_1, x_2, y_1, y_2)\right]\right)^{\frac{1}{p}}$$

By applying the Cauchy-Schwartz inequality, we have:

$\text{H2SW}_p(\mu, \nu; c, g_1, g_2)$

$$\leq \left( \mathbb{E}\left[ \inf_{\pi \in \Pi(\mu,\nu)} \int (\sqrt{\psi_1^2 + \psi_2^2})^p (\sqrt{(g_1(\theta_1, x_1) - g_1(\theta_1, y_1))^2 + (g_2(\theta_2, x_2) - g_2(\theta_2, y_2))^2})^p d\pi(x_1, x_2, y_1, y_2) \right] \right)^{\frac{1}{p}}$$

$$\leq \left( \mathbb{E}\left[ \inf_{\pi \in \Pi(\mu,\nu)} \int (|g_1(\theta_1, x_1) - g_1(\theta_1, y_1)| + |g_2(\theta_2, x_2) - g_2(\theta_2, y_2)|)^p d\pi(x_1, x_2, y_1, y_2) \right] \right)^{\frac{1}{p}}$$

$$\leq \left( \mathbb{E}\left[ \inf_{\pi \in \Pi(\mu,\nu)} \int |g_1(\theta_1, x_1) - g_1(\theta_1, y_1)|^p d\pi(x_1, x_2, y_1, y_2) \right] \right)^{\frac{1}{p}}$$

$$+ \left( \mathbb{E}\left[ \inf_{\pi \in \Pi(\mu,\nu)} \int |g_2(\theta_2, x_2) - g_2(\theta_2, y_2)|^p d\pi(x_1, x_2, y_1, y_2) \right] \right)^{\frac{1}{p}}$$

$$= \left( \mathbb{E}\left[ \inf_{\pi \in \Pi(\mu_1,\nu_1)} \int |g_1(\theta_1, x_1) - g_1(\theta_1, y_1)|^p d\pi(x_1, y_1) \right] \right)^{\frac{1}{p}}$$

$$+ \left( \mathbb{E}\left[ \inf_{\pi \in \Pi(\mu_2,\nu_2)} \int |g_2(\theta_2, x_2) - g_2(\theta_2, y_2)|^p d\pi(x_2, y_2) \right] \right)^{\frac{1}{p}}$$

$$= \text{GSW}_p(\mu_1, \nu_1; g_1, c) + \text{GSW}_p(\mu_2, \nu_2; g_2, c),$$

where the last inequality is due to the Minkowski's inequality.

(ii) From (i), we have $\text{H2SW}_p(\mu, \nu; c, g_1, g_2) \leq \text{GSW}_p(\mu_1, \nu_1; g_1, c) + \text{GSW}_p(\mu_2, \nu_2; g_2, c)$. When, $g_1, g_2$, and $c(x, y) = |x - y|$ are linear defining functions, we have:

$$\text{GSW}_p(\mu_1, \nu_1; g_1, c) = \left( \mathbb{E}\left[ \inf_{\pi \in \Pi(\mu_1,\nu_1)} \int (|\theta^\top x_1 - \theta^\top y_1|^p d\pi(x_1, y_1) \right] \right)^{\frac{1}{p}}$$

$$\leq \left( \mathbb{E}\left[ \inf_{\pi \in \Pi(\mu_1,\nu_1)} \int (\|\theta\|_2 \|x_1 - y_1\|_2^p d\pi(x_1, y_1) \right] \right)^{\frac{1}{p}}$$

$$\leq \left( \mathbb{E}\left[ \inf_{\pi \in \Pi(\mu_1,\nu_1)} \int (\|x_1 - y_1\|^p d\pi(x_1, y_1) \right] \right)^{\frac{1}{p}}$$

$$= \left( \inf_{\pi \in \Pi(\mu_1,\nu_1)} \int (\|x_1 - y_1\|^p d\pi(x_1, y_1) \right)^{\frac{1}{p}}$$

$$= W_p(\mu_1, \nu_1; c).$$

Similarly, we have $\text{GSW}_p(\mu_2, \nu_2; g_1, c) \leq W_p(\mu_2, \nu_2; c)$. Therefore, we obtain the proof of $\text{H2SW}_p(\mu, \nu; c, g_1, g_2) \leq W_p(\mu_1, \nu_1; c) + W_p(\mu_1, \nu_1; c)$.

(iii) When $g_1, g_2$ are linear defining functions, we have:

$\text{H2SW}_p(\mu, \nu; c, g_1, g_2)$

$$\leq \left( \mathbb{E}\left[ \inf_{\pi \in \Pi(\mu,\nu)} \int (|\theta_1^\top x_1 - \theta_1^\top y_1|) + |\theta_2^\top x_2 - \theta_2^\top y_2|)^p d\pi(x_1, x_2, y_1, y_2) \right] \right)^{\frac{1}{p}}$$

$$\leq \left( \mathbb{E}\left[ \inf_{\pi \in \Pi(\mu,\nu)} \int (|\theta_1^\top x_1 - \theta_1^\top y_1|) + |\theta_2^\top x_2 - \theta_2^\top y_2|)^p d\pi(x_1, x_2, y_1, y_2) \right] \right)^{\frac{1}{p}}$$

$$\leq \left( \mathbb{E}\left[ \inf_{\pi \in \Pi(\mu,\nu)} \int (|x_1 - y_1|) + |x_2 - y_2|)^p d\pi(x_1, x_2, y_1, y_2) \right] \right)^{\frac{1}{p}}$$

$$= \left( \inf_{\pi \in \Pi(\mu,\nu)} \int (|x_1 - y_1|) + |x_2 - y_2|)^p d\pi(x_1, x_2, y_1, y_2) \right)^{\frac{1}{p}}$$

When $p = 1$, we obtain:

$$\text{H2SW}_1(\mu, \nu; c, g_1, g_2) \leq \left( \inf_{\pi \in \Pi(\mu,\nu)} \int (|x_1 - y_1|) + |x_2 - y_2|) d\pi(x_1, x_2, y_1, y_2) \right)^{\frac{1}{p}}$$

$$= W_1(\mu, \nu; c, c),$$

which completes the proof.

## A.5 Proof of Proposition 4

Let $p \geq 1$, $c(x,y) = |x-y|$, $\mu \in \mathcal{P}(\mathbb{R})$ with the corresponding empirical distribution $\mu_n$, we assume that there exists $q > p$ such that the $q-$th order moment of $\mu$ i.e, $M_q(\mu) = \int_{\mathbb{R}} |x|^q d\mu(x)$, is bounded by $B < \infty$. From Theorem 1 in [20], there exists a constant $C_{p,q}$ such that:

$$\mathbb{E}\left[W_p^p(\mu_n, \mu; c)\right] \leq C_{p,q} B \begin{cases} n^{-1/2} \text{ if } q > 2p, \\ n^{-1/2} \log(n)^{\frac{1}{p}} \text{ if } q = 2p, \\ n^{-(q-p)/q} \text{ if } q \in (p, 2p). \end{cases}$$

We show that $\mathcal{HHR}_{\theta_1,\theta_2,\psi}^{g_1,g_2} \sharp \mu$ has finite bounded moments. In particular, we have:

$$\begin{aligned} M_k(\mathcal{HHR}_{\theta_1,\theta_2,\psi}^{g_1,g_2} \sharp \mu) &= \int_{\mathbb{R}} |t|^k d(\mathcal{HHR}_{\theta_1,\theta_2,\psi}^{g_1,g_2} \sharp \mu)(t) \\ &= \int_{\mathbb{R}^{d_1} \times \mathbb{R}^{d_2}} |\psi_1 g_1(\theta_1, x_1) + \psi_2 g_2(\theta_2, x_2)|^k d\mu(x_1, x_2) \\ &\leq \int_{\mathbb{R}^{d_1} \times \mathbb{R}^{d_2}} (\psi_1^2 + \psi_2^2)^{k/2} (g_1(\theta_1, x_1)^2 + g_2(\theta_2, x_2)^2)^{k/2} d\mu(x_1, x_2) \\ &\leq \int_{\mathbb{R}^{d_1} \times \mathbb{R}^{d_2}} (|g_1(\theta_1, x_1)| + |g_2(\theta_2, x_2)|)^k d\mu(x_1, x_2), \end{aligned}$$

where the first inequality is due to the Cauchy-Schwarz inequality and the second inequality is due to the fact that $\|x\|_2 \leq |x|$. For the linear defining functions $g(\theta, x) = \theta^\top x$, we have $|g(\theta, x)| = |\theta^\top x| \leq \|x\|_1$. For the circular defining functions $g(\theta, x) = \|x - r\theta\|_2 \leq \|x - r\theta\|_1 \leq \|x\|_1 + \|r\theta\|_1 \leq \|x\|_1 + r$. Therefore, we have:

$$\begin{aligned} M_k(\mathcal{HHR}_{\theta_1,\theta_2,\psi}^{g_1,g_2} \sharp \mu) &\leq \int_{\mathbb{R}^{d_1} \times \mathbb{R}^{d_2}} (|x_1| + |x_2| + C_{g_1,g_2})^k d\mu(x_1, x_2) \\ &= \int_{\mathbb{R}^{d_1} \times \mathbb{R}^{d_2}} \sum_{i=0}^{k} k^i (|x_1| + |x_2|)^i C_{g_1,g_2}^{k-i} d\mu(x_1, x_2) \\ &= \sum_{i=0}^{k} k^i C_{g_1,g_2}^{k-i} \int_{\mathbb{R}^{d_1} \times \mathbb{R}^{d_2}} (|x_1| + |x_2|)^i d\mu(x_1, x_2) \\ &\leq \sum_{i=0}^{k} k^i C_{g_1,g_2}^{k-i} M_i(\mu), \end{aligned}$$

where $C_{g_1,g_2} = 0$ if $g_1, g_2$ are linear, $C_{g_1,g_2} = r$ if $g_1$ and $g_2$ are linear and circular respectively (exchangeable), and $C_{g_1,g_2} = 2r$ if both $g_1$ and $g_2$ are circular.

Now, using the triangle inequality of H2SW (Theorem 1), we have:

$$\begin{aligned} \mathbb{E} \left| \text{H2SW}_p(\mu_n, \nu_n; c, g_1, g_2) - \text{H2SW}_p(\mu, \nu; c, g_1, g_2) \right| \\ \leq \mathbb{E} \left| \text{H2SW}_p(\mu, \mu_n; c, g_1, g_2) + \text{H2SW}_p(\nu, \nu_n; c, g_1, g_2) \right| \\ \leq \mathbb{E} \left| \text{H2SW}_p(\mu, \mu_n; c, g_1, g_2) \right| + \mathbb{E} \left| \text{H2SW}_p(\nu, \nu_n; c, g_1, g_2) \right| \\ \leq \left( \mathbb{E} \left| \text{H2SW}_p^p(\mu, \mu_n; c, g_1, g_2) \right| \right)^{\frac{1}{p}} + \left( \mathbb{E} \left| \text{H2SW}_p^p(\nu, \nu_n; c, g_1, g_2) \right| \right)^{\frac{1}{p}}, \end{aligned}$$

where the last inequality is due to Holder's inequality. Combining with previous results, we obtain:

$$\begin{aligned} \mathbb{E} \left| \text{H2SW}_p(\mu_n, \nu_n; c, g_1, g_2) - \text{H2SW}_p(\mu, \nu; c, g_1, g_2) \right| \\ \leq C_{p,q}^{\frac{1}{p}} \left( \sum_{i=0}^{q} q^i C_{g_1,g_2}^{q-i} (M_i(\mu) + M_i(\nu)) \right)^{\frac{1}{p}} \begin{cases} n^{-1/2p} \text{ if } q > 2p, \\ n^{-1/2p} \log(n)^{\frac{1}{p}} \text{ if } q = 2p, \\ n^{-(q-p)/pq} \text{ if } q \in (p, 2p), \end{cases} \end{aligned}$$

which completes the proof.

## A.6 Proof of Proposition 5

For any $p \geq 1$, and $\mu, \nu \in \mathcal{P}(\mathbb{R}^{d_1} \times \mathbb{R}^{d_2})$, using the Holder's inequality, we have:

$$\mathbb{E}|\widehat{\text{H2SW}}_p^p(\mu, \nu; c, g_1, g_2, L) - \text{H2SW}_p^p(\mu, \nu; c, g_1, g_2)|$$

$$\leq \left( \mathbb{E}|\widehat{\text{H2SW}}_p^p(\mu, \nu; c, g_1, g_2, L) - \text{H2SW}_p^p(\mu, \nu; c, g_1, g_2)|^2 \right)^{\frac{1}{2}}$$

$$= \left( \mathbb{E} \left| \frac{1}{L} \sum_{l=1}^{L} \text{W}_p^p(\mathcal{HHR}_{\theta_{1l},\theta_{2l},\psi_l}^{g_1,g_2} \sharp \mu, \mathcal{HHR}_{\theta_{1l},\theta_{2l},\psi_l}^{g_1,g_2} \sharp \nu; c) - \mathbb{E} \left[ \text{W}_p^p(\mathcal{HHR}_{\theta_1,\theta_2,\psi}^{g_1,g_2} \sharp \mu, \mathcal{HHR}_{\theta_1,\theta_2,\psi}^{g_1,g_2} \sharp \nu; c) \right] \right|^2 \right)^{\frac{1}{2}}$$

$$= \left( Var \left[ \frac{1}{L} \sum_{l=1}^{L} \text{W}_p^p(\mathcal{HHR}_{\theta_{1l},\theta_{2l},\psi_l}^{g_1,g_2} \sharp \mu, \mathcal{HHR}_{\theta_{1l},\theta_{2l},\psi_l}^{g_1,g_2} \sharp \nu; c) \right] \right)^{\frac{1}{2}}$$

$$= \frac{1}{\sqrt{L}} Var \left[ \text{W}_p^p(\mathcal{HHR}_{\theta_{1l},\theta_{2l},\psi_l}^{g_1,g_2} \sharp \mu, \mathcal{HHR}_{\theta_{1l},\theta_{2l},\psi_l}^{g_1,g_2} \sharp \nu; c) \right]^{\frac{1}{2}},$$

which completes the proof.

# B Additional Materials

**HHRT with more than two marginals.** We now extend the definition of HHRT to $K > 2$ mariginals.

**Definition 4** (Hierarchical Hybrid Radon Transform)**.** *Given $K \geq 2$, given defining functions $\{g_k : \mathbb{R}^{d_k} \times \Omega_i \to \mathbb{R}\}_{i=k}^{K}$, the Hierarchical Hybrid Radon Transform $\mathcal{HHR} : \mathbb{L}_1(\mathbb{R}^{d_1} \times \ldots \times \mathbb{R}^{d_K}) \to \mathbb{L}_1\left(\mathbb{R} \times \Omega_1 \ldots \times \Omega_K \times \mathbb{S}^{K-1}\right)$ is defined as:*

$$(\mathcal{HHR}f)(t, \theta_1, \ldots, \theta_K, \psi)$$

$$= \int_{\mathbb{R}^{d_1} \times \ldots \times \mathbb{R}^{d_K}} f(x_1, \ldots, x_K) \delta \left( t - \sum_{k=1}^{K} \psi_k g_k(x_k, \theta_k) \right) dx_1 \ldots dx_K. \quad (12)$$

**H2SW with more than two marginals.** From the new definition of HRRT on $K > 2$ marginals, we now can define H2SW between joint distributions with $K$ marginals.

**Definition 5.** *For $p \geq 1, K \geq 2$, defining functions $g_1, \ldots, g_K$, the hierarchical hybrid sliced Wasserstein-p (H2SW) distance between two distributions $\mu \in \mathcal{P}(\mathcal{X}_1 \times \ldots \times \mathcal{X}_K)$ and $\nu \in \mathcal{P}(\mathcal{Y}_1 \times \ldots \times \mathcal{Y}_K)$ with an one-dimensional ground metric $c : \mathbb{R} \times \mathbb{R} \to \mathbb{R}^+$ is defined as:*

$$H2SW_p^p(\mu, \nu; c, g_1, \ldots, g_K)$$

$$= \mathbb{E}_{(\theta_1, \ldots, \theta_K, \psi) \sim \mathcal{U}(\Omega_1 \times \ldots \times \Omega_K \times \mathbb{S}^{K-1})}[W_p^p(\mathcal{HHR}_{\theta_1, \ldots, \theta_K, \psi}^{g_1, \ldots, g_K} \sharp \mu, \mathcal{HHR}_{\theta_1, \ldots, \theta_K, \psi}^{g_1, \ldots, g_K} \sharp \nu; c)], \quad (13)$$

*where $\mathcal{HHR}_{\theta_1, \ldots, \theta_K, \psi}^{g_1, \ldots, g_K} \sharp \mu$ and $\mathcal{HHR}_{\theta_1, \ldots, \theta_K, \psi}^{g_1, \ldots, g_K} \sharp \nu$ are the one-dimensional push-forward distributions created by applying HHRT.*

**Lorentz Model and Busemann function.** The Lorentz model $\mathbb{L}^d \in \mathbb{R}^{d+1}$ of a d-dimensional hyperbolic space is [7]:

$$\mathbb{L}^d = \left\{ (x_1, \ldots, x_d) \in \mathbb{R}^{d+1}, -x_0 y_0 + \sum_{i=1}^{d} x_i y_i = -1, x_0 > 0 \right\}.$$

Given a direction $\theta \in T_{x_0} \mathbb{L}^d \cap \mathbb{S}^d$, $x \in \mathbb{L}^d$, the Busemann function is:

$$B(x, \theta) = \log(-\langle x, x_0 + \theta \rangle).$$

**Busemann function on product Hadamard manifolds.** For distributions supports on the product of $K \geq 2$ Hadamard manifolds with the corresponding Busemann functions $B_1, \ldots, B_K$, we have a Busemann function of the product manifolds is:

$$B(x_1, \ldots, x_K, \theta_1, \ldots, \theta_K) = \sum_{k=1}^{K} \lambda_k B_k(x_k, \theta_k),$$

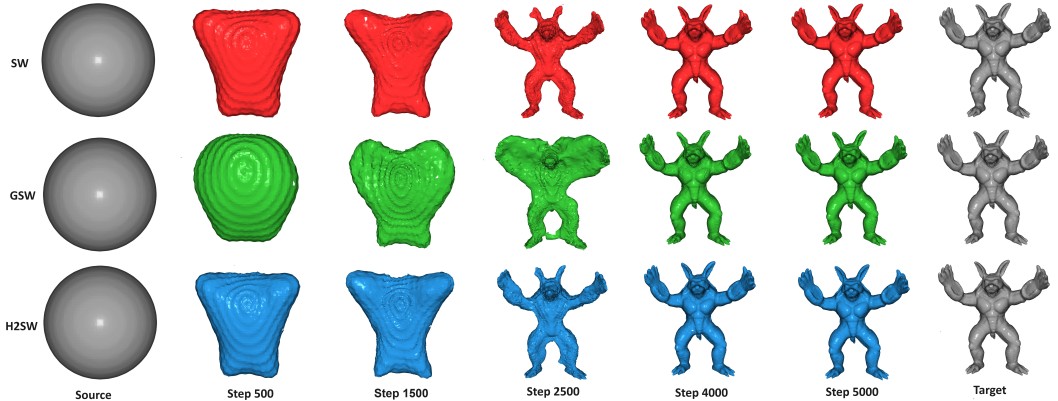

Figure 6: Visualization of deformation from the sphere mesh to the Armadillo mesh with $L = 100$.

for $(\lambda_1, \ldots, \lambda_K) \in \mathbb{S}^{K-1}$. The Cartan-Hyperbolic Sliced-Wasserstein distance use a fixed value of $(\lambda_1, \ldots, \lambda_K)$ e.g., $(\lambda_1, \ldots, \lambda_K) = (1/\sqrt{K}, \ldots, 1/\sqrt{K})$ (see [2]). In our proposed H2SW, we treat $(\lambda_1, \ldots, \lambda_K)$ as a random variable follows $\mathcal{U}(\mathbb{S}^{K-1})$ and the value of H2SW is defined as the mean of such random variable.

## C   Related Works

**HHRT and Generalized Radon Transform.** HHRT can be also seen as a special case of GRT [3] with the defining function $g(x, \theta) = \psi_1 g_1(x_1, \theta_1) + \psi_2 g_2(y, \theta_2)$ with $x = (x_1, x_2)$ and $\theta = (\theta_1, \theta_2, \psi)$ ($\Omega = \Omega_1 \times \Omega_2 \times \mathbb{S}$). However, without approaching via the hierarchical construction, the injectivity of the transform might be a challenge to obtain.

**HHRT and Hierarchical Radon Transform.** Hierarchical Radon Transform (HRT) [43] is the composition of Partial Radon Transform and Overparameterized Radon Transform, which is designed specifically for reducing projection complexity when using Monte Carlo estimation. Moreover, HRT is introduced with linear projection and does not focus on the problem of comparing heterogeneous joint distributions. In contrast to HRT, the proposed HHRT is the composition of multiple partial Generalized Radon Transform and Partial Random Transform, which is suitable for comparing heterogeneous joint distributions.

**HHRT and convolution slicers.** Convolution slicers [40] are introduced to project an image into a scalar. It can be viewed as a Hierarchical Partial Radon Transform i.e., small parts of the image are transformed first, then be aggregated later. Although convolution slicers can separate global and local information as HHRT, they focus on the domain of images only and have not been proven to be injective. Again, HHRT is designed to compare heterogeneous joint distributions and is proven to be injective in Proposition 2. As a result, H2SW is a valid metric while convolution sliced Wasserstein [40] is only a pseudo metric. Moreover, H2SW can also use convolution slicers when having marginal domains as images.

## D   Additional Experiments

**3D Mesh Deformation.** As mentioned in the main text, we present the deformation visualization to the Armadillo mesh with $L = 100$ in Figure 6, and the deformation visualization to the Stanford Bunny o mesh with $L = 10$ and $L = 100$ in Figure 3- 7 in turn. The quantitative result for the Armadillo mesh is given in Table 2. Here, we set the step size to 0.1. From these results, we see that the proposed H2SW gives the best flow deformation flow in general. The performance gap is especially larger when $L = 10$ i.e., having a small number of projections.

---

[2] https://github.com/clbonet/Sliced-Wasserstein_Distances_and_Flows_on_Cartan-Hadamard_Manifolds/blob/0eb05450e7f9f27586d0ddb1ce6e58f07eb75786/Experiments/xp_otdd/OTDD_SW.ipynb

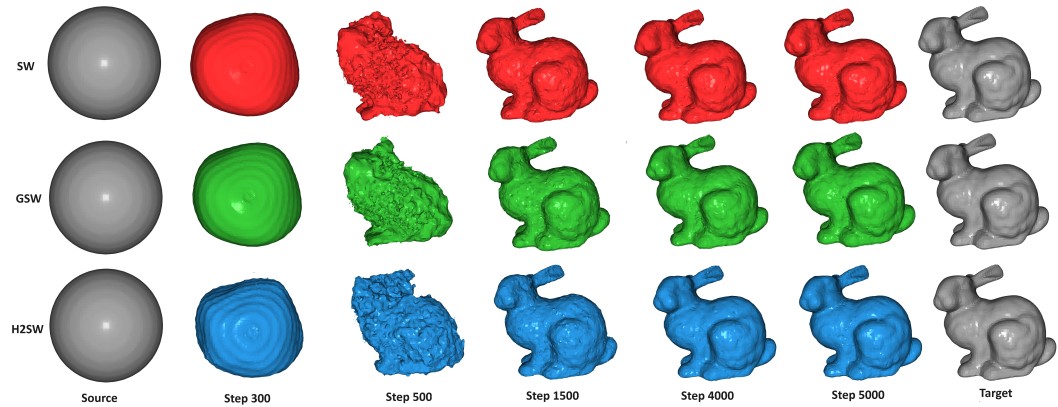

Figure 7: Visualization of deformation from the sphere mesh to the Stanford Bunny mesh with $L = 100$.

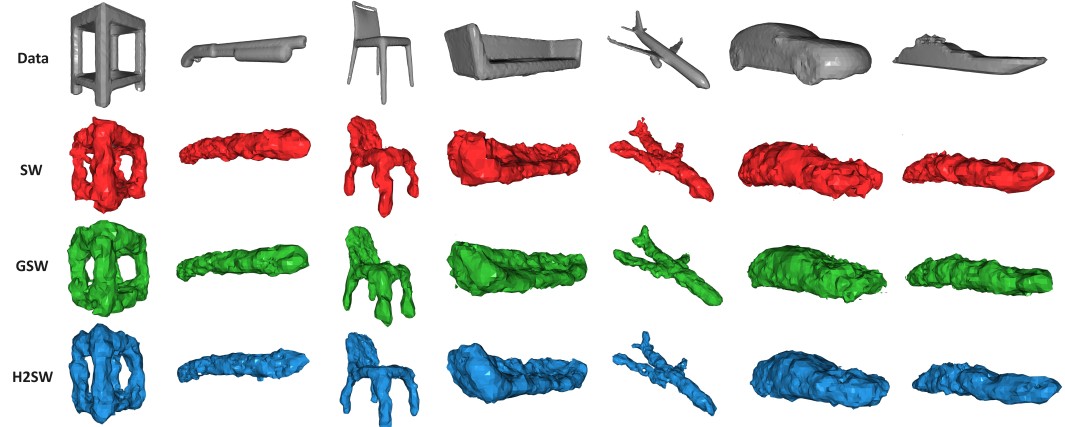

Figure 8: Visualization of some randomly selected reconstruction meshes from autoencoders trained by SW, GSW, and H2SW in turn with the number of projections $L = 100$ at epoch 500.

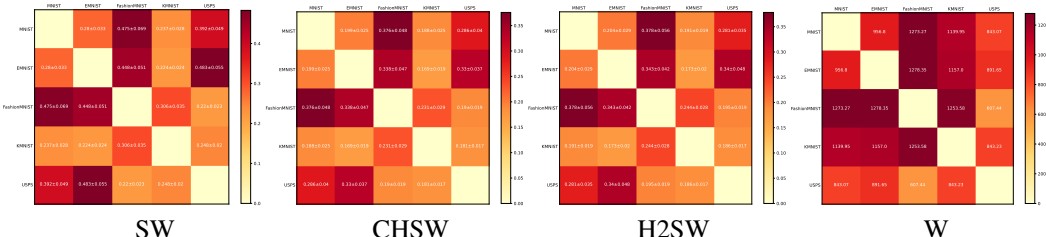

Figure 9: Cost matrices between datasets from SW, CHSW, and H2SW with $L = 100$.

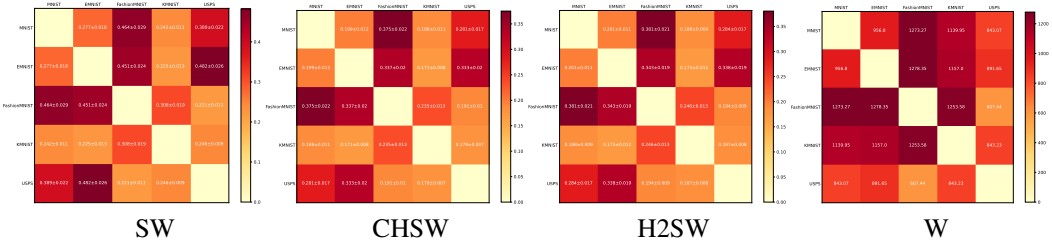

Figure 10: Cost matrices between datasets from SW, CHSW, and H2SW with $L = 500$.

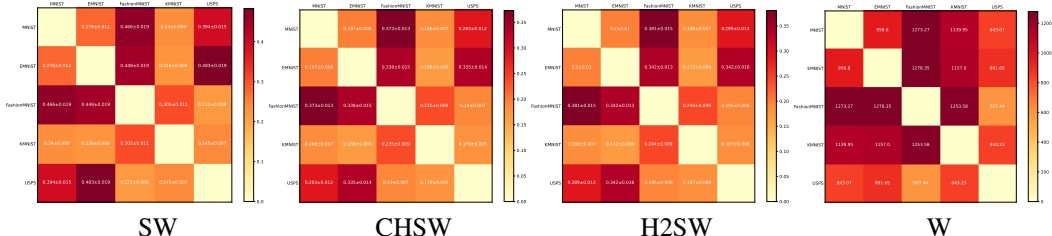

Figure 11: Cost matrices between datasets from SW, CHSW, and H2SW with $L = 1000$.

**Deep 3D mesh autoencoder.** We first report the neural network architectures that we use in the experiments.

- The encoder: Conv1d(6,64,1) → BatchNorm1d → LeakyReLU(0.2) → Conv1d(64, 128, 1) → BatchNorm1d → LeakyReLU(0.2) → Conv1d(128, 256, 1) → BatchNorm1d → LeakyReLU(0.2) → Conv1d(256, 512, 1) → BatchNorm1d → LeakyReLU(0.2) → Conv1d(512, 1024, 1) → BatchNorm1d → LeakyReLU(0.2) → Max-Pooling → Linear(1024, 1024).
- The decoder: Linear(1024, 1024) → BatchNorm1d → LeakyReLU(0.2) → Linear(1024, 2048) → BatchNorm1d → LeakyReLU(0.2) → Linear(2048, 4096) → BatchNorm1d → LeakyReLU(0.2) → Linear(2048, 2048*6). The output of the decoder is the concatenation of the location and normal vector. We normalize the normal vector to the unit-sphere.

As mentioned in the main text, we report the reconstruction of randomly selected meshes for $L = 100$ at epoch 500 in Figure 8. We see that the reconstructed meshes at epoch 500 are visually worse than the reconstructed meshes at epoch 2000. Therefore, the joint Wasserstein distances in Table 3 are consistent with the qualitative results.

**Dataset Comparison.** We follow the same procedure in Section 6.2 in [9]. We refer the reader to the reference for a detailed description. Here, we show the cross-dataset cost matrices with the number of projections $L = 100$ in Figure 9, $L = 500$ in Figure 10, and $L = 1000$ Figure 11.

# E Computational Infrastructure

For the non-deep-learning experiments, we use a HP Omen 25L desktop for conducting experiments. For 3D mesh autoencoder experiments, we use a single NVIDIA A100 GPU.

