# OpenReview forum: "Hierarchical Hybrid Sliced Wasserstein: A Scalable Metric for Heterogeneous Joint Distributions"
_NeurIPS.cc/2024/Conference — NeurIPS 2024 poster_

### Official Review · Reviewer_XX3o · 2024-06-29

**Soundness:** 4
**Presentation:** 4
**Contribution:** 3
**Rating:** 7
**Confidence:** 4

**Summary:**

This work focuses on introducing a Sliced-Wasserstein distance between heterogeneous joint distributions, i.e. distributions whose ambient space lies in a product space, with possibly different geometries for each space. This new distance relies on a new hierarchical hybrid Radon transform. The authors derive several theoretical properties of this new discrepancy, and apply it to three tasks: 3D mesh deformation and autoencoders whose underlying space can be seen as $\mathbb{R}^3 \times S^2$, and dataset comparisons, with datasets embedded on $\mathbb{R}^{d_1}\times \mathbb{L}^{d_2}$ where $\mathbb{L}$ is the Hyperbolic manifold.

**Strengths:**

This work introduces a new Sliced-Wasserstein distance to compare distributions on any product spaces. The relevance of this new distance is shown on different experiments, and many theoretical properties are derived.

- New distance to compare probability distributions on product spaces
- Theoretical properties such as conditions under which the proposed discrepancy is a distance, projection complexity, and connections with other distances
- Nice applications on 3D meshes and on dataset comparisons

**Weaknesses:**

To my opinion, there are few weaknesses. First, the applications on 3D meshes are nice and large scale (10000 points) but only in few dimensions. Moreover, on this application, the only distance used for the sphere component is the circular defining function. It would have been nice to compare it with e.g. a spherical Radon transform.

On the theoretical side, there are some parts which could be more clear, notably around Proposition 1 and 2. I give more details into the Questions section.

**Questions:**

In the statements of Proposition 1 and 2, it is stated "For some defining function g such as linear, circular, and homogeneous polynomials with an odd degree". I believe it works for defining functions which are injective? If it is the case, it would be more clear in my opinion to state, let $g$ be injective, and write somewhere else examples of injective defining functions.

Under Definition 2, it is written "The reason for using PRT for the final transform is that the previous PGRTs are assumed to be able to transform the non-linear structure to a linear line". Without reading the appendix and understanding that HHRT can be seen as a composition of PRT and PGRT, which is stated after, this sentence is not clear.

In Proposition 3, I guess $W_1(\mu,\nu,c,c)=W_1(\mu,\nu,c)$.

A missing reference on injective defining functions, which might give another example for which HHSW is a distance is [1].

Only 2 examples of heterogeneous data are given (3d meshes and datasets). Are there other examples?

Typos:
- Line 52: "In addition, a line of work...": misses verb
- In Definition 2, I do not think $\mathbb{S}$ is defined.
- In the proof of Proposition 3, in the equation after line 851: $\|x_1--y_1\|_2^p$
- Line 317: "By varying $L\in\{\dots\} For"
- Refs [20] and [21] are the same.
- Refs [56] and [57] are the same.



[1] Chen, Xiongjie, Yongxin Yang, and Yunpeng Li. "Augmented sliced Wasserstein distances." arXiv preprint arXiv:2006.08812 (2020).

**Limitations:**

Yes

---

> ### Author Rebuttal · Authors · 2024-08-04
>
> We would like to thank the reviewer for the time and constructive feedback.
>
> **Q17**. First, the applications on 3D meshes are nice and large scale (10000 points) but only in few dimensions ... It would have been nice to compare it with e.g. a spherical Radon transform.
>
> **A17**. For the low-dimensional comment, H2SW can still be used in any dimension,  however, we believe 3D data is still a very important data modality. As discussed in Example 1, we can use other types of transformation for the spherical domain including Funk-Radon Transform, Vertical Slice Transform, Parallel Slice Transform, Spherical Radon Transform, and Stereographic Spherical Radon Transform. We agree that using a spherical-type Radon Transform is more theoretically sounded than a circular Generalized Radon Transform. However, circular GRT is easier and more efficient in computation and implementation.
>
> We have replaced the circular GRT with the Stereographic Spherical Radon transform (SSRT) which is a spherical Radon Transform type, and run an additional experiment on deformation with the Armadillo mesh. We obtain the following result :
>
> | Distances | Step 100 ($W_{c_1,c_2}\downarrow$)  | Step 300 ($W_{c_1,c_2}\downarrow$) | Step 500 ($W_{c_1,c_2}\downarrow$) |
>  Step 1500 ($W_{c_1,c_2}\downarrow$) | Step 4000  ($W_{c_1,c_2}\downarrow$) | Step  5000 ($W_{c_1,c_2}\downarrow$) |
>
> |    H2SW  L=10|1840.73$\pm$1.282 | 1422.667$\pm$7.813| 1058.171$\pm$5.362 | 95.672$\pm$4.376 | 6.326$\pm$0.151 | 2.602$\pm$0.201 |
>
> |H2SWs L=10| 1840.574$\pm$4.505 | 1423.448$\pm$2.994 | 1061.584$\pm$3.9 | 95.618$\pm$3.044 | 6.365$\pm$0.106 | 2.595$\pm$0.038|
>
>   |  H2SW  L=100|1839.347$\pm$1.986 | 1417.1$\pm$3.677 | 1048.895$\pm$4.008 | 86.078$\pm$0.623 | 4.61$\pm$0.431 | 1.086$\pm$0.177|
>
> |H2SWs L=100| 1839.577$\pm$0.949 | 1417.785$\pm$1.502 | 1050.187$\pm$4.421 | 86.431$\pm$1.878 | 5.192$\pm$0.449 | 1.085$\pm$0.191|
>
> From the result, we can see that the result of SSRT is similar to using circular GRT. However, it is worth noting that SSRT does not have hyperparameters while circular GRT has one hyperparameter. Hence,  SSRT might be a good alternative for circular GRT when we want to avoid hyperparameter tuning.
>
> [9] Stereographic Spherical Sliced Wasserstein Distances, Tran et al.
>
> **Q18**. In the statements of Proposition 1 and 2, it is stated "For some defining function g such as linear, circular, and homogeneous polynomials with an odd degree". I believe it works for defining functions which are injective? If it is the case, it would be more clear in my opinion to state, let $g$ be injective, and write somewhere else examples of injective defining functions.
>
> **A18**.  Thank you for the insightful comment. For the defining function, it is not necessary to be injective in order to make the GRT injective. For example, the linear defining function $g(x,\theta)=\langle x,\theta \rangle$ is not injective i.e., if $\theta$ are orthogonal to $x_1$ and $x_2$ with $x_1 \neq x_2$, we still have $g(x_1,\theta)=g(x_2,\theta)=0$. However, by considering all possible values of $\theta$, the GRT is still injective.
>
> We assume that the reviewer is referring to spatial Radon Transform [7] where an injective function can be used to transform the original support $x$, then the linear projection is applied. In HHRT, we have not used such a function to transform the original support. However, it is possible to extend HHRT into spatial HHRT i.e., using different injective functions to transform marginal arguments first, then apply generalized one-dimensional projection after that. By doing so, we can make the generalized one-dimensional projection easier to separate two distributions as discussed in [10]. The only challenge is to design meaningful injective functions on manifolds and special domains. We will include this discussion in the revision of the paper and leave this promising investigation to future work.
>
> [10] Augmented Sliced Wasserstein Distances, Chen et al.
>
> **Q19**. Under Definition 2, it is written "The reason for using PRT for the final transform is that the previous PGRTs are assumed to be able to transform the non-linear structure to a linear line". Without reading the appendix and understanding that HHRT can be seen as a composition of PRT and PGRT, which is stated after, this sentence is not clear.
>
> **A19**. Thank you for your comments.  We will add an equation to the main text as a rigorous explanation along with the sense.
>
> **Q20**. On Typos
>
> **A20**. Thank you for pointing out the typos. We will fix them in the revision.
>
> **Q21**. A missing reference on injective defining functions, which might give another example for which HHSW is a distance is [1].
>
> **A21**. Thank you for your reference. We will include the paper in the revision of the paper. As discussed in **Q18**, we believe that the augmented approach is a promising extension for H2SW. We will leave this extension to future work.
>
> **Q22**. Only 2 examples of heterogeneous data are given (3d meshes and datasets). Are there other examples?
>
> **A22**. Thank you for your great questions. We would like to give a few other examples of heterogeneous distributions.  The first example is distributions over features and class labels in domain adaptation [7].  The second example is distributions over data variables and latent variables in joint contrastive inference [8]. The third example is point-clouds with additional attributes e.g., colors, opacity, and so on.  In addition, distributions over multimodal data (e.g., text and image) can also be seen as heterogeneous joint distributions since the geodesic metric on the text space and the geodesic metric on the image space are different. However, the challenge is that for such complex data, the geodesic metrics are unknown which requires additional effort for metric learning.
>
> [7] Joint distribution optimal transportation for domain adaptation, Courty et al
>
> [8] Wasserstein Variational Inference, Ambrogioni et al

---

> > ### Comment · Reviewer_XX3o · 2024-08-08
> >
> > Thank you for answering my questions.
> >
> > **A 17.** Thank you for adding this experiment, which seems promising.
> >
> > **A 18.** Sorry, I was confused about the injectivity. Indeed, the defining function does not need to be injective. If I understand well the proof of Proposition 1, the injectivity holds as the associated generalized Radon transform is injective. Thus, maybe it would be better to assume this, and provide besides examples of defining functions satisfying this property?
> >
> > Given the answer of the authors, and revisions such as bigger Figures (in particular Figure 1 but also Figure 4 on which we cannot read the values), I will improve my score to 7.

---

> ### Author Response · Authors · 2024-08-08
> **Response to Reviewer XX3o**
>
> Thank you for your quick reply,
>
> On **A18**, you are right, for the proof of the metricity of the distance, we only need the injectivity of the transform (not the defining function). It seems that we restrict the paper a little bit to a very well-known set of Radon Transform with some particular well-investigated defining function. We will revise the Proposition 1 based on your suggestions. Again, we would like to say thank for your constructive feedback. Please feel free to ask if you have any questions!
>
> Best regards,

---

### Official Review · Reviewer_p3fW · 2024-07-03

**Soundness:** 3
**Presentation:** 3
**Contribution:** 3
**Rating:** 8
**Confidence:** 4

**Summary:**

The paper proposes a new metric for comparing heterogeneous joint distributions, called Hierarchical Hybrid Sliced Wasserstein (H2SW). The proposed metric is tested for shape deformation, for training shape autoencoder and comparing datasets.
The proposed metric is based on a new slicing operator (Hierarchical Hybrid Radon Transform, HHRT).

**Strengths:**

The paper's contributions are in defining a novel slicing operator (HHRT) and its associated metric (H2SW).

The paper is clear, well written and interesting to read.

**Weaknesses:**

The third contribution listed (lines 81-82) is confusing (and in my opinion should be rephrased/tone down): the paper ignores  other attempt(s) for deforming shapes using heterogenous joint distributions using likewise point clouds combined with normals - e.g. see https://doi.org/10.1016/j.patcog.2018.02.021 for an example with 2D and 3D shapes using L2 metric between heterogenous distributions.

Standard error (e.g. tab 3) are too large to rank/compare tested methods with confidence.

Experiments in section 4.3 are not easy to grasp ( rely on reader to read recent [9])

Minor remarks:

- reference [56] == [57] ?

- [50] published 10.1088/1361-6420/acf156 but arxiv cited

- line 49: "interested distributions"  ?

- line 85: "Hadamand"

- line 89: "distance s"

- Fig 1 too small fonts

- line 248 "the injectivity of the Buseman projection has not been known at the moment" : do you mean no math proof available yet?

- line 251-252: "H2SW is a generic distance for heterogeneous joint distributions in  which marginal domains do not necessary manifolds":  "do not necessary manifolds"  meaning??

- line 259-60 "we compare H2SW with SW and  Cartan-Hadamard Sliced-Wasserstein (CHSW) in comparing datasets on the product of Hadamard  manifolds" : repetition compare/comparing ... sentence unclear

- Fig 4 too small fonts

- line 909 "Hierarchial"

- line 918 "Armadill mesh"

- Fig 9/10/11 too small font

- Buseman projection -> Busemann projection ?

**Questions:**

Can authors clarify the application 4.3 ? i.e. what is the data and it is encoded ($\mathbb{L}^{d_2}$ ?)?

**Limitations:**

The work presented focuses on contribution to defining an  OT metric however it may be that HHRT has applications in other areas (?)

---

> ### Author Rebuttal · Authors · 2024-08-04
>
> We would like to thank the reviewer for insightful comments and constructive feedback. We would like to answer questions from the reviewers as follow:
>
> **Q11**. The third contribution listed (lines 81-82) is confusing (and in my opinion should be rephrased/tone down): the paper ignores other attempt(s) for deforming shapes using heterogenous joint distributions using likewise point clouds combined with normals - e.g. see https://doi.org/10.1016/j.patcog.2018.02.021 for an example with 2D and 3D shapes using L2 metric between heterogenous distributions.
>
> **A11**.  Thank you for your insightful comment. We will adjust the writing to tone down our claim. We would like to clarify that we do not claim that we are the first that utilize point clouds combined with normals for representing 3D shapes.  The main contribution of the paper is the proposal of a hierarchical hybrid sliced Wasserstein (H2SW) which is a new variant scalable of sliced Wasserstein for heterogenous joint distributions. In greater detail, the time complexity of H2SW is $\mathcal{O}(n \log n)$ and the space complexity is $\mathcal{O}(n)$ for $n$ is the number of supports (the number of points of the 3D shape). Moreover, H2SW leverages optimal transport which retains the geometric structure. In contrast, L2 distance requires evaluating density at all supports of two distributions which makes the time complexity and space complexity $\mathcal{O}(n^2)$. In addition, to create smooth transitions, the L2 distance relies on KDE methods which are not necessary for H2SW. We will include the mentioned paper in the reference and add the discussion to the revision.
>
> **Q12**. Standard error (e.g. tab 3) are too large to rank/compare tested methods with confidence.
>
> **A12**. Thank you for the comments. Since SW variants are approximated with the Monte Carlo estimation with $L>0$ projecting directions. Increasing the number of projections $L$ can help to reduce the randomness (the standard error) as demonstrated in Table 3. Due to the limitation of the computational infrastructure, we could not increase $L$ further to obtain results in an acceptable time. Nevertheless, we believe H2SW can show its benefits when $L=2000$ since both the mean and the standard error are lower. Moreover, compared to SW, H2SW is better considerably which demonstrates the benefit of the hierarchical hybrid Radon Transform.
>
> **Q13**. Experiments in section 4.3 are not easy to grasp ( rely on reader to read recent [9])
>
> **A13**. Thank you for your comment. We refer the reader to **Q15** for a detailed discussion.  We will add this discussion to the revision of the paper.
>
> **Q14**. Minor remarks and typos.
>
> **A14**. Thank you for pointing out. We will fix typos and revise the paper based on your suggestions.
>
> **Q15**. Can authors clarify the application 4.3 ? i.e. what is the data and it is encoded ($\mathbb{L}^{d_2}$?)?
>
> **A15**.  In  Section 4.3, we want to compare datasets as discussed in [5] and follow the approach in [6]. In particular, a dataset consists of samples with features and labels. For example, the MNIST dataset has images of digits with associated labels (the class index of the digit). For each class, we want to embed to $\mathbb{L}^{d_2}$. To do that, we represent each class as an empirical distribution of images and compute the pair-wise Wasserstein distance between class distributions to obtain a pair-wise cost matrix between classes from two datasets. We then use multidimensional scaling to obtain the embedding of classes on $\mathbb{L}^{d_2}$. Finally, we represent a dataset as a set of samples on $\mathbb{R}^{d_1} \times \mathbb{L}^{d_2}$ with $\mathbb{R}^{d_1}$ is for the image and $ \mathbb{L}^{d_2}$ for the class embedding. We will add this detail to the revision of the paper.
>
> [5] Geometric Dataset Distances via Optimal Transport, Alvarez-Melis et al.
>
> [6] Sliced-Wasserstein Distances and Flows on Cartan-Hadamard Manifolds, Bonet et al.
>
> **Q16**. The work presented focuses on contribution to defining an OT metric however it may be that HHRT has applications in other areas (?)
>
> **A16**. Thank you for your insightful questions. From the perspective of signal processing, HHRT can be used as a feature extractor [7] for other tasks like Radon Transform e.g., image classification tasks [8]. In this case, the feature from HHRT can be used to classify 3D shapes or datasets as discussed in the paper. Moreover, we believe that HHRT can be also used for data denoising i.e., transforming noisy data and then reconstructing a less noisy version with inverse transforms.
>
> [7] Data representation with optimal transport, Martin et al
>
> [8] Radon Cumulative Distribution Transform Subspace Modeling for Image Classification, Shifat-E-Rabb et al.

---

> > ### Comment · Reviewer_p3fW · 2024-08-10
> > **Thank you for the responses provided**
> >
> > Thanks for the information provided. I am happy with rebuttal answers.

---

> > > ### Author Response · Authors · 2024-08-10
> > > **Response to Reviewer p3fW**
> > >
> > > We would like to say thank you again for your time and constructive suggestions. We will revise the paper based on the discussion with the reviewers. Please feel free to raise questions if you have one.
> > >
> > > Best regards,

---

### Official Review · Reviewer_deuA · 2024-07-09

**Soundness:** 2
**Presentation:** 1
**Contribution:** 2
**Rating:** 6
**Confidence:** 4

**Summary:**

Wasserstein distance measures the distance between two distributions and is particularly important in machine learning, especially in generative models. However, computing the Wasserstein distance is computationally intensive. For homogeneous distributions, Sliced Wasserstein (SW) and Generalized Sliced Wasserstein (GSW) significantly reduce time and memory complexity. However, while they are computationally friendly for heterogeneous distributions, they fail to capture the underlying heterogeneity of the spaces. This paper addresses these issues with the following contributions:
1.Novel Slicing Operator: The paper proposes a novel slicing operator for heterogeneous joint distributions, called the Hierarchical Hybrid Radon Transform (HHRT).

2. Hierarchical Hybrid Sliced Wasserstein (H2SW): Using HHRT, the authors develop the Hierarchical Hybrid Sliced Wasserstein (H2SW) distance tailored for heterogeneous distributions.

3.Empirical Validation: The superiority of HHRT is demonstrated on heterogeneous distribution datasets.

4.Extension of Radon Transform: The partial Radon Transform is extended to the partial generalized Radon Transform (PGRT), and the monotonicity of PGRT is proven, thereby confirming the monotonicity of HHRT.

5. Comprehensive Analysis: The paper investigates the topological, statistical, and computational properties of H2SW.

**Strengths:**

1.1Pioneering Metric for Heterogeneous Joint Distributions: The paper introduces Hierarchical Hybrid Sliced Wasserstein (H2SW), a novel metric specifically designed for comparing heterogeneous joint distributions, a task that conventional methods struggle with.

1.2A Novel Slicing Operator for Heterogeneous Data: To address the challenges of comparing heterogeneous data, the paper proposes a novel slicing operator for heterogeneous joint distributions, named Hierarchical Hybrid Radon Transform (HHRT).

1.3 Extending Partial Radon Transform: The paper extends the concept of Partial Radon Transform (PRT) to the more general Partial Generalized Radon Transform (PGRT), expanding its applicability.

1.4. Unveiling Properties and Connections:

a.  Monotonicity of HHRT and PGRT: The paper delves into the properties of HHRT and PGRT, establishing their monotonicity, a crucial aspect for ensuring meaningful comparisons.

b. Unveiling the Link between HHRT and PGRT: The paper explores the intricate connection between HHRT and PGRT, shedding light on their underlying relationship.

c. A comprehensive analysis of H2SW is conducted, encompassing its topological properties, statistical properties, and computational properties.

**Weaknesses:**

The article lacks attention to detail in several areas.
The paper needs to clearly explain heterogeneous joint distributions,
such as by detailing why 3D shape deformation is considered a heterogeneous joint distribution.
• P22-849: The explicit expression for GSWp p(u, v; g, c) is not immediately obvious. A detailed derivation from the definition should be provided.
• The proof of Proposition 1 seems to be simplified. By fixing y, Let
fy
(x) = f(x, y), then PGRf degenerates to GRf. So f1,y(x) = f2,y(x) ⇒
f1(x, y) = f2(x, y).
• Page-5: In Proposition 4, what is the specific meaning of the subscripts
n for µn, vn? How do they differ from and relate to µ1 in Proposition 3?
What is the connection between the definition of P
q and the parameter
q?
• Here are some writing issues or areas that need further elaboration:
– Page-3,100: L1 should be clarified as the Lebesgue integrable function space.
– Page-3,110:The full name of CDF :Cumulative Distribution Function.
– Page-4,157: The functions f1, f2 should belong to the function
space L1(Rd1 × Rd1).
– Pqge-4,162: The variable x should be subscripted as x1,i, x2,i.
– Page-20,812-: The equation is not balanced on both sides. The
left side should be rewritten in the form of the right side.
– Page-21,840/844: redundant left parenthesis
– Page-22,851-: there is an extra ’-’ sign in the first inequality.
– Page-22,854-: The θ1, θ2 symbol should be moved outside the absolute value symbols.
– Page-22,855-: for p = 1, 1
p
should be removed

**Questions:**

see the weakness.

**Limitations:**

yes

---

> ### Author Rebuttal · Authors · 2024-08-04
>
> We would like to express our gratitude for the time and feedback from the reviewer. We would like to answer questions from the reviewers as follows:
>
> **Q6**. The paper needs to clearly explain heterogeneous joint distributions, such as by detailing why 3D shape deformation is considered a heterogeneous joint distribution
>
> **A6**. Thank you for your comments. It is worth noting that the notion of heterogeneous joint distributions arises only when considering the metric spaces of marginals, which are not usually accounted for in conventional probabilistic models. We define heterogeneous joint distributions as joint distributions that have marginals supported on different domains with different supported marginal ground metrics. For example, $\mu(x,y) \in \mathcal{P}(\mathbb{R}^d \times \mathbb{S}^{d-1})$ is a  heterogeneous joint distribution. The reason is that $\mathbb{R}^d$ supports the Euclidean metric $c_1(x_1,x_2)=\|x_1-x_2\|$ as the geodesic while $\mathbb{R}^d$ supports the great circle metric $c_2(y_1,y_2)=arccos(\langle y_1,y_2\rangle)$. The heterogeneity comes from the difference in geodesic distances for each subset of arguments.
>
> 3D shapes are often presented as triangle meshes, which contain vertices and faces. Vertices are location vectors in 3-dimensional real space ($\mathbb{R}^3$) while faces are sets of 3 vertices that are pairwise connected. In their original form, 3D shapes are expensive to store and difficult to manipulate. As a result, implicit representations have been proposed, one of which is the shapes-as-points representation [3]. In this representation, a 3D shape is represented by a set of points that lie on the surface of the 3D shape (constructed manually or by sampling). Additionally, for each point, the associated normal vector—the unit directional orthogonal vector to the surface at the point—is stored. As a result, a 3D shape is represented as a set of points on the product space $\mathbb{R}^3 \times \mathbb{S}^2$. This representation is lightweight, suitable for neural network processing on 3D meshes, and can be converted back to the triangle mesh representation using Poisson surface reconstruction.
>
>
> To convert a set of points to a probability distribution, we can assign a weight to each point (the summation of all weights is 1) to turn the set of points into a discrete distribution. Since points are on the product space $\mathbb{R}^3 \times \mathbb{S}^2$, the resulting discrete distribution is a heterogeneous joint distribution as discussed above. In the experiments, we assign uniform weight to points i.e., turning the set of points into an empirical distribution.
>
> [3] Shape As Points: A Differentiable Poisson Solver, Peng et al.
>
> **Q7**. P22-849: The explicit expression for $GSW_ p(\mu, \nu; g, c)$ is not immediately obvious. A detailed derivation from the definition should be provided.
>
> **A7**. Thank you for your comments. Since the measures $\mu$ and $\nu$ are supported on an open set and the defining function $g$ is continuous to the support of $\mu$, the defining function $g$ is measurable. By the definition of the push-forward measure and change of variable formula [4], $\int   |g(\theta,x)-g(\theta,y)|^{p}d\pi (x,y) = \int   |x’-y’|^{p}d\pi’ (x’ ,y’) = $ where $\pi’$ is a joint distribution of $\mathcal{GR}^g_\theta \sharp \mu$ and $\mathcal{GR}^g_\theta \sharp \nu$ by pushing forward $\pi$ through $g(\theta,x)$ and $g(\theta,y)$.  As a result, we obtain $W_p^p(\mathcal{GR}^g_\theta  \mu,\mathcal{GR}^g_\theta  \sharp \nu;c)$ as defined in Equation 4 in the paper. We will add a more detailed derivation to the revision of the paper.
>
> [4] https://en.wikipedia.org/wiki/Pushforward_measure
>
> **Q8**. The proof of Proposition 1 seems to be simplified. By fixing y, Let $f_y (x) = f(x, y)$, then PGRf degenerates to GRf. So $f_{1,y}(x) = f_{2,y}(x) \Rightarrow f_1(x, y) = f_2(x, y)$.
>
> **A8.** Thank you for your detailed comments. We will include this derivation in the revision of the paper.
>
> **Q9**. Page-5: In Proposition 4, what is the specific meaning of the subscripts n for $\mu_n$, $\nu_n$? How do they differ from and relate to µ1 in Proposition 3? What is the connection between the definition of $\mathcal{P}_q$ and the parameter $q$?
>
> **A9**. Thank you for pointing this out. Here, $\mu_n$ and $\nu_n$ are empirical distributions over $n$ random samples from $\mu$ and $\nu$ respectively. In particular,  let $X_1,\ldots,X_n \overset{i.i.d}{\sim} \mu$, $\mu_n = \frac{1}{n} \sum_{i=1}^n \delta(x-X_i)$. It seems that this notation overlaps with the marginal distribution notation $\mu_1$ in Proposition 3. We will change the notation from $\mu_n$ to $\hat{\mu}_n$ in the revision.  For the definition of $\mathcal{P}_q$, it is the set of all probability distributions with finite $q$-th moments ($q\in \{1,2,3,\ldots,...\}$). We will add a paragraph to explain notations in the revision of the paper.
>
> **Q10**.  On writing issues or areas that need further elaboration:
>
> **A10**. Thank you for pointing out. We will fix all writing issues in the revision of the paper.

---

### Official Review · Reviewer_E3oU · 2024-07-11

**Soundness:** 3
**Presentation:** 3
**Contribution:** 3
**Rating:** 6
**Confidence:** 3

**Summary:**

To address the issue that Sliced Wasserstein (SW) and Generalized Sliced Wasserstein (GSW) GSW are only defined between distributions supported on a homogeneous domain and using SW and GSW directly on the joint domains cannot make a meaningful comparison, this paper first extends the partial Radon Transform to the Partial Generalized Radon Transform (PGRT) to inject non-linearity into local transformation, and then proposes a novel slicing operator for heterogeneous joint distributions, named Hierarchical Hybrid Radon Transform (HHRT). The authors conduct experiments on optimization-based 3D mesh deformation and deep 3D mesh autoencoder to show the favorable performance of H2SW compared to SW and GSW. In general, this paper is well structured and presented.

**Strengths:**

1. This paper addressed the issue of that Sliced Wasserstein (SW) and Generalized Sliced Wasserstein (GSW) GSW are only defined between distributions supported on a homogeneous domain, and using SW and GSW directly on the joint domains cannot make a meaningful comparison.
2. This paper is well structured.

**Weaknesses:**

1. The authors did not clarify the main challenges of designing a sliced Wasserstein variant for heterogeneous joint distributions in the Introduction section. Additionally, they failed to explain how the proposed H2SW addresses these challenges. I suggest that the authors elaborate on these points to provide a clearer understanding of the significance and innovation of this work.
2. The paper involves many similar concepts, such as PRT, GRT, and PGRT. What are the relationships between them? Is the proposed PGRT a combination of PRT and GRT? These questions are confusing. I suggest that the authors provide a detailed explanation of the differences and connections between these concepts.
3. The text in Figure 1 is too small to read clearly, and the main idea it aims to convey is not clear enough. I suggest that the authors add some elements to make the figure easier to understand.
4. The definitions in the paper are sometimes too concise, lacking detailed explanations and context. For instance, when introducing new transforms and metrics, more background information and examples would help readers understand the practical applications of these definitions.
5. Some descriptions in the paper are overly verbose and the language lacks precision, which affects readability and may cause confusion. I recommend that the authors refine the writing to make it more concise and precise.

**Questions:**

Please clarify the main challenges of designing a sliced Wasserstein variant for heterogeneous joint distributions in the Introduction section, and how the proposed H2SW addresses these challenges.

**Limitations:**

The authors have addressed the limitations of their work.

---

> ### Author Rebuttal · Authors · 2024-08-04
>
> We would like to thank the reviewer for the time and the feedback.
>
> **Q1**. The authors did not clarify the main challenges of designing a sliced Wasserstein variant for heterogeneous joint distributions in the Introduction section. Additionally, they failed to explain how the proposed H2SW addresses these challenges.
>
> **A1**.  As discussed in the introduction, heterogeneous joint distributions are joint distributions that have marginals supported on different domains with different supported ground metrics. As a result, the joint ground metric between two points in the joint spaces is often a weighted combination of marginal ground metrics between corresponding marginal dimensions. For example, if the marginal domains are manifolds, the weighted combination of ground metrics is the geodesic ground metric on the joint space [1].
>
> To design a SW variant for heterogeneous joint distributions, we need to map probability distributions on the joint space to one-dimensional distributions on the real line to apply the closed form of the one-dimensional Wasserstein distance with a strictly convex-based ground metric. A meaningful map should be injective, computationally efficient, and geometrically meaningful. Injectivity is essential for preserving the identity of the indiscernible property of the sliced metric; i.e., when all one-dimensional projections are equal, the original distributions are also equal. Moreover, the map requires fast computation to maintain the near-linear computation time of one-dimensional Wasserstein distances. The geometric meaning is necessary for preserving the topological structure of the original space of distributions under the Wasserstein distance with the geodesic ground metric to some extent. In more detail, the composition of the map and the chosen strictly convex-based ground metric in one dimension create a pull-back metric on the original joint space. The pull-back metric will not account for the difference between the two marginal domains if we use conventional maps from the Radon Transform and the Generalized Radon Transform. As a result, the geometry of the original joint space is lost.
>
> To address these challenges, the proposed H2SW utilizes the hierarchical hybrid Radon Transform (HHRT), which is the composition of the Partial Radon Transform and multiple domain-specific Partial Generalized Radon Transforms. Intuitively, HHRT first projects marginal domains onto real lines and then projects the joint space of those lines onto the final real line. By doing so, HHRT can retain more of the geometric structure of the original joint space. When marginal domains are Hadamard Manifolds and marginal Busemann projections are used, the HHRT is equivalent to using Busemann projections (not unique) on the joint space [2]. We would like to emphasize that HHRT cannot entirely preserve the original geometry; however, it is the first attempt to tackle the challenge since it is injective and has fast computation with efficient choices of marginal generalized Radon Transforms.
>
> [1] Learning Mixed-Curvature Representations in Product Spaces, Gu et al.
>
> [2] Metric spaces of non-positive curvature, Bridson et al.
>
> **Q2**. The paper involves many similar concepts, such as PRT, GRT, and PGRT. What are the relationships between them? Is the proposed PGRT a combination of PRT and GRT?
>
> **A2**. We would like to clarify the relationships of PRT, GRT, and PGRT as follows:
>
> First, the RT maps a function over $\mathbb{R}^d$ to a function over $\mathbb{S}^{d-1} \times \mathbb{R}$ by integrating over hyperplanes (Equation 2). Given a projecting direction $\theta  \in \mathbb{S}^{d-1}$, the RT provide a  linear map  from $\mathbb{R}^d$ to $\mathbb{R}$. However, a linear map is not geometrically correct for some domains such as manifolds in $\mathbb{R}^d$. Therefore, the generalized Radon Transform is introduced by changing from integrating over hyperplanes to integrating over hypersurfaces (controlled by the defining function - line 116). PRT  is also an extension of RT which transforms only a subset of arguments via integrating over hyperplane on the corresponding sub-space of the arguments. PRT is useful in signal-processing problems where we are particularly interested in only a subset of arguments due to some prior knowledge. As a natural extension, the proposed  Partial Generalized Radon Transform (PGRT) generalizes PRT by changing from integrating over the hyperplanes on the corresponding sub-space to integrating over the hypersurfaces. Overall, PGRT serves as a building block of HHRT since it is used to transform marginal arguments of the joint spaces. Since the marginal spaces are heterogeneous, different types of PRGT are used (with different defining functions).
>
> **Q3**. The text in Figure 1 is too small to read clearly, and the main idea it aims to convey is not clear enough.
>
> **A3**. We will increase the font sizes and revise the figures accordingly in the revision.
>
> **Q4**. The definitions in the paper are sometimes too concise, lacking detailed explanations and context. For instance, when introducing new transforms and metrics, more background information and examples would help readers understand the practical applications of these definitions.
>
> **A4**. We will add the background information of the previous transforms to the revision as discussed in **Q2**. In the paper, we also present Example 1, which discusses the usage of HHRT in the application of a 3D mesh with shape as points representation. We will add more detail on how HHRT is applied step by step and the computation of H2SW in this example in the revision. Additionally, we will include a new example comparing datasets using Busemann projections for a more comprehensive discussion.
>
> **Q5**. Some descriptions in the paper are overly verbose and the language lacks precision, which affects readability and may cause confusion.
>
> **A5**. We will adjust the writing in the revision based on your suggestions.

---

> > ### Comment · Reviewer_deuA · 2024-08-12
> > **thanks for the response**
> >
> > Thank you for your response. After considering your feedback, I have decided to maintain my score.

---

### Author Rebuttal · Authors · 2024-08-04

Dear chairs and reviewers,

First, we would like to thank the reviewers for spending time reviewing our paper and providing constructive feedback. We would like to summarize some additional results and common questions from the reviewers:

* As suggested by Reviewer **XX3o**, we added an experiment where we replaced the circular generalized Radon Transform with a spherical-based Radon Transform, i.e., Stereographic Spherical Radon Transform (SSRT), for the hierarchical Hybrid Radon Transform. We observed that the SSRT gives comparable results while having no hyperparameters. Overall, we would like to emphasize that HHRT can utilize any type of domain-specific transforms, which have been active research areas recently.

* As suggested by Reviewer **E3oU** and **deuA**, we will add a more detailed discussion on heterogeneous joint distributions in the revised paper. In summary, the notion of heterogeneous joint distributions appears when we consider the metric space of supports. We would like to refer to a short description in the paper: “Heterogeneous joint distributions are joint distributions that have marginals supported on different domains with different supported ground metrics.” Overall,  we can treat a heterogeneous joint distribution as a special multivariate distribution, which has different structures for some blocked marginal distributions.

* We will revise the paper by fixing the typos, adding a detailed discussion, improving figures, and providing more detailed proofs as suggested by the reviewers.

Best regards,

Authors

---

### Decision · Program_Chairs · 2024-09-25

**Decision:**

Accept (poster)

**Comment:**

There is agreement among the reviewers that this paper contributes novel ideas and new concepts to an important topic and addresses a real need with potentially high impact on the ML research community.   The paper needs a careful round of editing by the authors prior to the final submission.